# Arranged Marriages in Multilateral Partnerships—Investigating Sustainable Human Development Financing of Belize in the World Bank Group: A Brand Relationship Theory Approach

**Jacqueline D. Ifield** * and **Chia-Han Yang**

Institute of Creative Industries Design, National Cheng Kung University, Tainan City 701, Taiwan
* Correspondence: jacquelinedifield@gmail.com

**Abstract:** The structure of multilateral financial partnerships has many relationship challenges, which need to be solved to positively impact sustainable human development. There is a lack of understanding in the development relationship between the so-called developed and developing countries, and development economics theories and research, which guide policies, knowledge, and funding to nations in need. Amid widespread pleas for change to the structure, Belize is a country, which remains in an economic development crisis 41 years after joining the World Bank Group. This original paper, uniquely positions "World Bank" as a brand, and adds to missing empirical research on Belize and development economics with a mixed-methods, brand relationship approach. The researchers perform a survey of 20 years of Belize government personnel dialogues about the Bank, and apply Fournier's (1998) brand relationship theories as tools to measure their perceptions of the Bank as well as get a deeper understanding of the relationship. This investigative research finds that Belize perceives it has an "arranged marriage" type brand relationship with the Bank: it is not a sustainable development partner. This brand relationship also negatively affects the way government personnel see themselves and their abilities. The World Bank Group must innovate its development economics methods and practices, assert its social mission, and meet the development needs of its members by first building genuine brand bonds with them. Debtor member countries must re-define their worth, join together, and design their own paths to sustainable development. All countries are developing.

**Keywords:** sustainable development; development economics; brand; branding; multilateral framework; human development; Belize; World Bank; knowledge economy

## 1. Introduction

Development of human capital through academics has become very important as more developed countries advance to knowledge economies [1–5]. It is a group of knowledge-based industries, which focuses on ideas instead of physical capital. Robertson [6] (p. 6) says they started when countries such as the United Kingdom and the United States were searching for new ways of making money during the "1970's oil shocks" and "crisis of capitalism." They produced, "new boundaries, geometries and temporalities, and open[ed] the way for renewed economic growth" [6] (p. 6). The knowledge economy includes communications, research, and the unlimited possibilities of creating "new science and technological knowledge at the global technological frontier as well as absorbing and distributing existing knowledge" to solve national and international, real-life problems [1]. Although it has proved difficult to measure [6], back in 2008, the knowledge economy was estimated at 10% percent annual growth with at least 7% of global GDP by the United Nations (UN) [7,8]. Quality education with lifelong learning, which is a characteristic of the knowledge economy [9,10], is also Goal #4 of the 17 Sustainable Development Goals (SDG) [11,12] in the United Nations' 2030 Agenda for Sustainable Development [13]. Citizens must be the

agents of economic change in knowledge [1]. The SDGs give more importance and urgency to empower people with knowledge. Leaders of struggling nations want to unlock opportunities for their citizens in this global economic knowledge revolution [1,14–16], and do so sustainably, but securing sustainable multilateral financing to develop and improve their higher education sector has many relationship challenges [10,17–19].

The scientific literature shows that many multilateral financial challenges are deeply rooted in relationship gaps between the so-called developing and developed countries [10,18]. This has created misunderstanding in the theories and research of development economics, which guide development policies, knowledge, and funding [3,17,20,21]. Strategically, the World Bank Group is positioned in development economics to play a very critical close-the-gap role in the development funding relationship between creditor and debtor nations as it tries to end extreme poverty, and aid in financing the fulfillment of United Nations projects. Some researchers applaud the Bank's work and projects [22,23]. Others seek to explore the theories and reasons behind its policies [22,24–30]. But, the banking cooperative has faced a lot of criticism from people of developing countries it is supposed to help [9,31,32], and others, who question its work and methods [3,6,9,10,18]. The Bank's loyalty between the two categorizations of countries has also been scrutinized [6,17,24,33].

A major part of the Bank's mission is to forge a sustainable end to extreme poverty in "developing countries" with financial products and knowledge services. Belize, for example, is a named developing country, which is in dire economic crisis, now 41 years after joining the Bank. According to the International Monetary Fund (IMF), WBG's big sister organization, the on-going COVID-19 pandemic has already brought a devastating blow to Belize's economy, which is marked by "a deep recession" [34]. The country "faces difficult challenges, including unsustainable public debt, widening external imbalances, and vulnerability to natural disasters and climate change" [34]. The restoration of debt sustainability to strengthen the currency peg remains a priority even though a 9.8% GDP growth was recorded in 2021 [35]. Belize is called, "a modest performer" in knowledge infrastructure with "university-industry collaboration in R&D" seen as an area of improvement [36].

Belize's Prime Minister Juan "John" Briceño, in his first speech to the United Nations General Assembly on 24 September 2021, made a bold plea for the reform and repurposing of the multilateral framework on sustainable development [37]. He pointed out failures in the development relationship between developed and developing countries: inequalities in COVID-19 vaccine distribution; funding models and methodologies, which he believes are inappropriate for countries such as his; GDP losses in developing nations caused by the climate crisis for which developed countries take no ownership; Taiwan's imperative inclusion in the membership; and the debilitating US embargo against Cuba. When PM Briceño returned in 2022 he was, "more pessimistic . . . more cynical and disappointed" in "no collective will" among leaders to update the multilateral system, mainly due to its focus on "fiscal orthodoxy" [38]. He called for a new global financial architecture and complained about the dismal 2022 update to the already failing SDGs, which he says is being ignored by the international financial institutions and multilateral development banks [38]. PM Briceño claimed that the World Bank loaned USD 99 billion in funds to developing countries in 2021, while in fact they need USD 4.3 trillion in financing [38]. Additionally, Barbados PM Mia Mottley asked for longer repayment terms [39]. PM Mottley let it be known that certain European countries had special concessions and one hundred years to repay financing; now, developing countries only have seven to ten years [39]. Leaders from across the world made similar pleas and recommendations for change [38–46]. A few days later, while a guest speaker at Stanford University, World Bank Group President, David Malpass, quantified the problem and said, loans to developing nations need "bank equity capitalization" because they are risky [47]. Malpass continued to say that these loans are made with what is left from global capital after loans to advanced nations, which have zero risks [47]. This is exactly the problem.

This research offers an original, brand relationship approach to both measure and understand the needs of Belize, as a debtor member country of the multilateral institution. It is

significant to development economics research in various ways. While researchers have focused on the surface level multilateral funding relationship in projects [3,6,9,10,17,18,31–33], theories, and methodologies [20,21,48–51], no one has tried to measure and define it from a brand perspective. It is also about the demand-side with the debtors, which Fine [52] says is missing in the literature. Additionally, although Teferra [18] asserts that most of the on-going scholarly debate on the banking cooperative does not question the perceived integrity of the Bank's findings, or knowledge content, no one has framed the World Bank Group as a brand with financial products, services, knowledge content, global indices, people, and experiences, which form a complex network of bonds and associated brand relationships with country members. This includes their governments, researchers and citizens. The researchers broaden Fournier's [53] brand relationship frameworks and theories and apply them as tools to measure and characterize the relationship, as well as diagnose its strength with a joint qualitative and quantitative approach. This research also exposes a lack of theory or framework on member-brand relationships, especially of countries in multilateral cooperatives. This is more complex and multifaceted than Fournier's [53] consumer-brand dyad. And while there is scientific evidence of the Bank's works in Africa [3,17], and Asian countries, such as Indonesia [10], Vietnam [14], China, and India [4], as well as Latin America [23], nothing is known in the literature about the Bank's work in Belize. Additionally, nothing is known about the nature of that relationship and its impact on sustainable human capital development within the small, "thinly populated" [54] (p. 11) [55] (p. 15), and data-scarce country. Almendarez [2] points to data, which show that Belize is lacking human capital and says it is needed, but he does not focus on its financing. Waight [56] makes a plea for strengthening the relationship between human capital and business in Belize, but she does not talk about human development funding challenges. Villanueva [57] uses the World Bank's influence to prop up her rationale for quality assurance in higher education in the country, but she does not acknowledge that the Bank has historically impeded the growth of this sector in developing countries [10,18]. This research is timely. The researchers are motivated by the important, urgent, and repeated calls for change to the multilateral funding system by Belize [37,38], and governments across the world [39,40,42–46]. The research gives answers that the World Bank Group must consider to build genuine partnerships and provide these client countries with the sustainable development financing they need.

This research aims to define the Government of Belize's brand relationship, and its quality [53], with the World Bank Group, as well as describe how that relationship impacts the country's sustainable development of the Belizean human capital for the knowledge economy. The research questions are: 1. What are the characteristics of the funding relationship of Belize's membership in the World Bank Group? 2. What is the brand relationship form [53] of Belize's membership in the World Bank Group, based on the perceptions of the Bank by Government of Belize personnel? 3. How does the brand relationship define challenges in meeting the country's sustainable development goals, especially for human development? The answers to the first question provide some background of the relationship.

It is useful to note the following definitions. The United Nations [58] says that sustainable development is "development that meets the needs of the present without compromising the ability of future generations to meet their own needs". Economic growth, social inclusion, and environmental protection are seen as paramount to ensuring it [58]. The results show that current needs are not being met, and the 2030 Sustainability Agenda did not have plans for the setbacks of a possible global crisis. A brand can be seen as, "a promise that is made to customers" [59], but, in this paper, a brand "has no objective existence at all: it is simply a collection of perceptions held in the mind of the consumer [client and customer]" [53] (p. 345). Brand relationships are, "what consumers do with brands to add meaning in their lives" [53] (p. 367). There are many interpretations and classifications of human capital; it is "productive wealth embodied in labor, skills and knowledge" [60], in addition to health [61]. There is now a focus on knowledge. The human capital the-

ory [62–66] sees things done to improve ourselves, such as getting a formal education, as economically driven investments [67], which have the promise of future financial gain and social mobility [19,68]. The knowledge economy is a type of economy that is driven by knowledge human capital, and particularly on the possibility and ability of a population's knowledge transformation into the economic value [69] (pp. 189–190). The knowledge economy places importance on higher education, research, and lifelong learning [9,10].

This paper starts with a relevant review of the literature, then a detailed account of the methodology. The results establish the facts concerning Belize's funding relationship with the World Bank Group, followed by answers to brand relationship type/form [53] and challenges to reach sustainable human development based on the brand relationship. A discussion ensues based on Fournier's [53] theory on brand relationship quality. Relevant conclusions are drawn on this brand relationship research in sustainable human development with recommendations for the World Bank Group, and decision makers of countries, such as Belize.

### 1.1. Country Relationship Challenges in Development Economics

Development economics started after the decolonization of formerly free European territories when the question of what to do with these newly independent, but poor, nations emerged at the end of the Cold War [51]. During this time, their poverty became more visible to more advanced countries [48]. Inikori [70] says, it started when students from these poor nations saw how inadequate the economics tools were for their own situation. Mechanical or neoclassical ideas from the experiences of developed-country economics were used on poor countries before the field evolved [48,51]. Wilber and Francis [49] say Economist Albert O. Hirschman criticized it because he recognized that the contexts are not the same. The 1979 Nobel Memorial Prize in Economic Sciences recipient W. Arthur Lewis, from the small island-state of St. Lucia, is credited with much of the foundations of development economics. Lewis' development economics ideas borrowed from the Arts of the classic economics and focused on pragmatism to meet policy needs. This approach shifted development economics thought away from the positivism of science and Keynes' ideas on the need for administration of economics [48]. Wisman [48] also adds that the co-recipient of the 1979 Nobel Memorial Prize in Economic Sciences, Theodore William Schultz, from the United States, aimed to disprove Lewis' work as non-scientific. There was an apparent battle in economics thinking from the arts to science, but Lewis's work partially brought it back to the arts [48].

In a paper, which focused on the theoretical and practical challenges of development economics, Alacevich [51] says, "old development economics" were "excessively theoretical for policy-oriented organizations and excessively empirical and incapable of producing elegant models for the academic world . . . " Olken [21] documents another revolution in the field. He adds that the methodologies in development economics have also had developed country-developing country relationship challenges, which demanded their evolution from macroeconomics to microeconomics, from growth models to development models, and from "at the frontier" thinking to "catch-up to the frontier" ideas. These changes were perceived as more relevant to developing countries. Most importantly, the field required a better understanding of the real situation in developing countries by shifting importance away from quantitative to qualitative data [20]. The advent of micro-empirical evidence from randomized control trials (RCT) became revolutionary [21]. However, Alacevich [51] found that conducting randomized control trials is tedious and it has limitation challenges.

However, as the field attempts to understand such questions as why some nations are poor [21], many researchers fail to acknowledge the conjoined history of the European, American, and African continents, for example, and what Inikori [70] calls the Atlantic system. History records show that European colonialists rendered them poor by overtaking their lands, killing their indigenous people, and repopulating the colonies with enslaved Africans, and others, who were de-humanized, indoctrinated, and stripped of their cultures, birthright, and potential. The wealth found and created by exploiting the physical and

labor capital in the Trans-Atlantic Slave Trade among these continents, and African Slavery on the American continent, which includes Belize and the Caribbean, was subsequently extracted for Western European, North American, and Australian development in industrial capitalism [70]. It was maintained with what Collins and Rhoades [17] call, "protectionist practices," which may be true even today. Alacevich [50] remarks that this question of what to do with poor nations paralleled the reorientation of the International Bank for Reconstruction and Development into the modern World Bank Group.

Alacevich [50,51] claims that "development economics was increasingly marginalized by the larger field of economics—with paradoxical results", and got reabsorbed into mainstream economics, but Lin [20] and Olken [21] disagree. Lin [20] proposes a third theoretical generation of the field as a "response" because of what he calls, the failure of the first- and second-generation development thinking, respectively called structuralism and neoliberalism. He highlights the ongoing struggle in the relationship between developed countries, which have the theories for development, and the developing countries, which "want" it. The former always tells the latter what to do to achieve modernization and industrialization.

Lin [20] says that structuralism advice in the multilateral framework of development economics is routinely based on asking developing countries to invest in infrastructure development. Neoliberalism theory asks for privatization of government assets to reduce public spending [20]. He also cites evidence of countries such as China and Singapore, which achieved success, even though they did not follow structuralism and neoliberalism advice [20]. Much of this advice and the funds to do it came from multilateral development organizations, such as the World Bank Group and the International Monetary Fund.

*1.2. World Bank Branding and Its Country Relationships*

The World Bank Group (WBG), as well as the IMF and the World Trade Organization (WTO), started in 1944, with the signing of the Bretton Woods agreement by representatives of forty-four countries. It was named the International Bank for Reconstruction and Development (IBRD), and was established to help in the reconstruction of European countries, which suffered damages in World War II. Riley [28] concludes that the Bank's funding relationship with Italy (1948–1951) helped to change its mission toward underdeveloped nations. During this transition, the Bank changed its mission from reconstruction to development and closed its economics department for a few years because they believed it was not necessary [51]. This brought a shift from economics policy to applied economics, and its work from long-term goals to short-term country projects, which were qualified based on a country's "creditworthiness" and "absorptive capacity" [51]. However, Inikori [70] views long-term development goals as more important for these countries. Bentley [29] (p. 201) believes that this period of World Bank Group changes was defined by a series of nexuses: a "neoclassical growth imperative . . . [which] marginalized direct poverty alleviation and legitimated growth" in the Bank's operations.

WBG now has 189 members and calls itself, "one of the world's largest sources of funding and knowledge for developing countries" [71]. The membership cooperative provides a huge array of financial products, services, and knowledge in content marketing products to these country clients in five institutions: International Bank for Reconstruction and Development (IBRD), International Development Association (IDA), International Finance Corporation (IFC), Multilateral Investment Guarantee Agency (MIGA), and the International Center for Settlement of Disputes (ICSID). IBRD and IDA are also grouped together as "The" World Bank. WBG's core values are stated as "impact", "integrity", "respect", "innovation", and "teamwork", while its mission is "reducing poverty, increasing shared prosperity, and promoting sustainable development" [71]. This is especially important in helping to finance the fulfillment of the United Nations Sustainable Development Goals as the multilateral development bank to members of the UN.

Bebbington, Guggenheim, Olson, and Woolcock [25] (pp. 39–40) inform us that development thoughts within the Bank have changed around different themes, from

"engineering and physical infrastructure", to "poverty", and "policy reform". Alacevich also notes an early focus on, "financing projects for infrastructures and directly productive activities" [50] (p. 652). Lending to governments was also a change in policy [24]. The Bank's definition of poverty has also seen changes [27,31].

### 1.3. World Bank Brand and Country Relationship Challenges

Collectively, "The" World Bank, and the World Bank Group are called, "not a typical financial institution" [18], "a predominantly economic institution" [50], and a bureaucracy [31]. It is also referenced as one of the "world policy organizations" [16], "global economic forces" [57], and an "international regulatory institution" [33]. Bresser-Pereira [24] expressed concerns about whether it is private or public, and questioned if it is turning into a commercial bank. He even contemplated the concept of a "development bank" [24]. The Bank is also named, "one of the world's most influential global" institutions in health [72] and knowledge: a "knowledge bank" or "the knowledge bank" [6,10,17,18,33].

Many researches have also questioned where the banking group's true loyalty lies [6,17,31,33]. Is the Bank's loyalty with its rich, majority-shareholder, creditor, developed country-clients, which steer its work [24,73], or poorer minority-shareholder, debtor, developing economies it is supposed to serve [9,31,32]? The Bank has been blasted for aligning power to developed nations, such as the United States and the United Kingdom [17]. Although Abbakumova [74] (p. 32) says that the World Bank Group (WBG) and the International Monetary Fund (IMF) work together to "ensure the stability of the international monetary and financial system," Bresser-Pereira [24] provides more information. Bresser-Pereira [24] (pp. 227–228) makes it known that the WBG and the IMF became executors of a debt power system in the 1980s, which includes "the [United States] Treasury and the [United States] Federal Reserve, and as consulting groups, the finance ministers of the G-7 and the chairmen (around 20) of the major commercial banks." This may be why membership in the IMF is a prerequisite to membership in the World Bank Group. Bresser-Pereira says, during the Regan Administration in the United States, the Bank faced an identity crisis and had no choice but to take on the additional role of promoting "privatization, liberalization and financial reform," "to manage the debt crisis and protect commercial banks" of creditor countries [24] (pp. 227–228). At the same time, it limited funding to only short-term projects in developing countries [24]. The Bretton Woods agreement became secondary [24]. Therefore, although the Bank's website only markets to developing countries, which implies less developed countries in need of funding, it is actually serving the two groups of countries: debtors and creditors.

### 1.4. Country Relationship Challenges in World Bank Branded Knowledge Project

Just over 20 years ago, the Task Force on Higher Education and Society [75], which was convened by UNESCO and the World Bank, stated that developing countries were homes to 80% of the world's population, but, due to lack of access, only about half of the world's higher education students live in these nations. They saw them as potential human capital and a valuable resource for a global knowledge economy (GKE). They gave developing countries a mandate: create (knowledge) human capital at home, and you will "exploit" benefits from national development, and participation in a global knowledge economy (GKE). The (World Bank/UNESCO) Task Force added that these countries risk increased poverty, "[global] intellectual and economic marginalization and isolation" from the "system" if they do not do it [75] (p. 18), but when they stated, "higher education is no longer a luxury . . . " [75] (p. 14), many in developing countries understood it as a monumental change in the World Bank's education policy [10,18,76]. This is because the World Bank Group had previously called higher education in developing countries indeed "a luxury", and refused to fund it or encourage others to do so. The Bank claimed that the rate on investment was not worth it even though there was growing demand for the sector [4,18]. It was later revealed that they used a faulty ROI assessment tool. The fact that the Bank never truly apologized left a "credibility gap" because of what should

be believed [18]. Additionally, the World Bank has also been accused of disrespecting countries' cultures to achieve its own goals [10,14].

Twenty years later, the concept of the global knowledge economy has yet to take shape and many developing countries have not created their own knowledge economies. But, according to the literature, the concept of the "global knowledge economy" was actually an imaginary [6,10], or a largely impossible concept [33]. It was facilitated and promoted [10,33] by the World Bank and others to birth KEs in developing countries with desired developed countries' international development and education policies, and without giving relevant details [6]. Roberts [33] proposed that it would align countries based on core and periphery, where people of developing countries would only be on the sidelines, watching real benefits in core countries, such as the United States. GKE policies were grounded in the second-generation development economics principle called neoliberalism [19,33].

By 2009, Roberts [33] concluded that the World Bank failed to end poverty with knowledge. But during these decades the World Bank has won the fight of development by education against UNESCO with an economics brand of human capital theory instead of UNESCO's social cultural theories [19]. The banking cooperative has seemingly narrowed (or expanded) its focus to the Human Capital Project, which tracks education and health [61].

The review of the literature shows that there are real relationship issues and misunderstandings between branded developing and developed countries on development. As much as science aims to be objective, these misunderstandings are negatively affecting theories and research in development economics. This situation leads to problems in projects by the World Bank Group because of the way its administrators create knowledge, draft policies, and assess the need for funding. The structure of the World Bank Group and differences in its branding and brand are also negatively impacted in the power struggle between these two groupings of countries. The dysfunctional system causes failure to meet development needs of those countries, which actually need the help. Juma and Agwara [15] (p. 232) say, "Development partnership has to be truly open and collaborative. Conventional judgments about project 'failure' and 'success' must be replaced with a greater emphasis on lessons learned. As Einstein put it, 'Anyone who has never made a mistake has never tried anything new'".

## 2. Materials and Methods

This is a mixed method research, which uses both quantitative and qualitative data to provide surface and deeper understanding of the complex brand relationship issue in development economics. The use of both types of methods also helps to validate the results. Lin [20] verifies that qualitative studies are a new trend and more appropriate in development economics than quantitative approaches, which were popular in its history. The researchers use guidelines of a systematic review of the literature [77] to avoid bias and to collect data from credible online sources of scientific and grey literature. A total of 172 archived and transcribed Belize news stories were retrieved from https://Channel5Belize.com (accessed on 30 October 2021) after a survey search of the term "world bank," from January 2000 to October 2021. Each mention was coded in an Excel spreadsheet after sampling to improve the table and coding. Eighty-eight "world bank" text mentions were extracted from dialogs in interviews and speeches made by Government of Belize (GOB) personnel for the brand relationship part of this paper. In a second round of data collection and coding, these 88 mentions expanded to 210 after "world bank" pronouns, products, and services were further isolated from the same news reports. The coding was based on categorical themes taken from Fournier's [53] brand relationship findings: initiation, emotions, bonds, symmetry, exclusivity, and rewards.

### 2.1. Data Selection

Belize and the World Bank Group are very relevant to this research. Belize was chosen because it is a member of the multilateral bank and there is a lack of research on the country.

The country also has environmental, economic, social, and human status problems in need of improvement. The People's United Party (PUP) and the United Democratic Party (UDP) are the two main political parties in Belize. Within the 2000–2021 data collection time span, which marks the time since the 2000 global knowledge economy mandate [75] was published, the PUP was in government from 2000 to February 2008 and November 2020 to present, while the UDP administered from February 2008 to November 2020. A profile of the Government of Belize (GOB) personnel data sample in Table A1 of Appendix A shows that the results are mostly informed by the UDP.

The World Bank Group was selected for this research because of its dual role as a multilateral development bank and its work on promoting knowledge economy, human capital, and the SDGs. It is also a brand with its own brand book. Due to concerns and evidence about the Bank's work in capital markets [10,24], instead of the social, it is also fitting to apply brand theories to its products, services, and experiences and find out how its members feel about them. A profile of the descriptions of "World Bank" mentions from the news data gathered is provided in Table A2 of Appendix B.

### 2.2. Data Analysis

Set questions and answer choices were used for accuracy to code the 88 and 210 data samples based on categories adapted from Fournier [53]: initiation, emotions, bonds, symmetry, exclusivity, and rewards. The "initiation" question is, "What initiates or causes brand mentions/contacts?" The "emotions" question is, "What emotions are conveyed in brand mentions/contacts towards World Bank Group?" The "bonds" question is, "What brand bond to World Bank Group relationship is enforced or reflected the most by speaker in brand mentions/contacts?" The "symmetry" question is, "Is the project's funding responsibility or work shared between GOB and World Bank Group?" The "exclusivity" question is, "Is World Bank Group the only funding agency mentioned or not?" The "rewards" question is, "What rewards does the Government of Belize experience in the relationship with the World Bank Group based on brand mentions/contacts?" The list of answers for emotions was taken from Plutchik's Wheel of Emotions [78]. These added more range to Fournier's two dimensions.

The coding results were compared with the characteristics of Fournier's [53] fifteen defined brand relationship forms to determine the brand relationship types/form of this case. Additionally, 108 of the 210 data were coded based on subject matter related to the 17 United Nations Sustainable Development Goals (SDG). A total of 108 samples were studied because they were taken from 2016 when the SDG started. The coded SDG data was used for cross analysis with the rewards data to obtain the real impact that WB has made in fulfilling these goals in Belize. Fournier's [53] theories on brand relationship forms and brand relationship quality aid in further analysis of the results and discussion of this research. Results were extracted from the main Excel spreadsheet into pivot tables and graphs for evidence and further analysis.

### 2.3. Justification

The dialog from interviews and speeches provides rich, first-hand information and context about the Government of Belize's brand perceptions of the cooperative banking group. The researchers posit that the 20-plus-year collection of news stories establishes the relationship between the World Bank Group and Belize, and in real time, which provides excellent, factual data for analysis of the Government of Belize's "lived experiences with their brands" [53]. News 5 at Great Belize Television's https://Channel5Belize.com (accessed on 30 October 2021) proved reliable because of the website's user-friendly search function and the company's award-winning, good reputation. The results from coding the 88 data samples and the 210 data samples had similar findings. This comparison, based on two rounds of data collection and coding, provides validity and accuracy to the results in this research. The results of the funding relationship of question one and brand relationship form of question two are also complimentary. This finding also provides validity to the results of

this research. Dissertations and theses used in the review of the literature are treated like white literature [79]. Grey literature is used because it is an essential supplementary source of data [80], which is especially vital in the absence of scientific literature on Belize. The researchers aimed to investigate the member relationship in this case, and found Fournier's theories appropriate because they are at the interpersonal level. An analysis of brand is important [81]; brands help in product/service differentiation as well as forming genuine relationships. Quality is better than loyalty in consumer–brand relationships [53].

## 3. Results

### 3.1. Belize and the World Bank's Funding Relationship

Belize became a member of the World Bank Group six months after its independence from Britain based on uncontested majority votes by the institutions' governors [82]. A World Bank press release states that the country's membership to the World Bank, the International Development Association (IDA), and the International Finance Corporation (IFC) was initially granted on 18 March 1982 [82]. The World Bank Group now classifies Belize as an "upper-middle income country" [83] based on its increased Gross Domestic Product (GDP). This categorization changed the country's membership from IDA to the International Bank for Reconstruction and Development (IBRD) and gave the country access to IBRD loans. Belize is also a shareholder in the WBG, but its decision-making power in the five institutions is currently minute: 0.05% of IBRD votes, 0.07% of IDA votes, 0.03% of IFC votes, and 0.15% of MIGA votes [84]. This contrasts sharply with the United States, for example, at 15.66% of IBRD votes, 9.86% of IDA votes, 18.79% of IFC votes, and 14.81% of MIGA votes [85]. The decision-making power of votes is mostly tied to money and the share capital of member countries in each institution [86].

In 2009, Prime Minister Dean Barrow of the United Democratic Party (UDP) called the World Bank, "the premiere international financial institution to this country" [87]. The World Bank Group's website page on Belize's loans and projects [88] publicizes that, up to 2021, Belize had eleven approved IBRD project loans with the Bank. These include the first in 1983: P006093—Road Maintenance and Rehabilitation Project (1983), P006094—Power Development Project (1986), P006097—Agriculture Credit and Export Development Project (1988), P006098—Road Maintenance and Rehabilitation Project 2 (1988), P006101—Primary Education Development Project (1991), P006104—Belize City Infrastructure Project (1993), P006103—Power Sector Development Project 2 (1994), P039292—Social Investment Fund (1997 and 2001), P040150—Roads and Municipal Drainage Project (2000), P111928—Municipal Development (2010), and P127338—Climate Resilient Infrastructure (2014) [88]. These loans totaled USD 131.2 million in Bank commitments as of 31 July 2021 [88]. Nine of the eleven listed projects were marked repaid [88].

However, a seemingly incomplete downloadable Excel source file (WB_Projects_1_500_downloaded_11_17_2021.xlsx), from a link on that page [88], shows that Belize asked the Bank for funding for 29 projects by 16 November 2021. The file is presented in part in Table 1. It shows that twenty loan projects are closed, four are dropped, four are in the pipeline, while one, called "Energy Resilience for Climate Adaptation," is active. There are no dates given for when the applications were sent/received. The "pipeline" and "dropped" projects have no approval dates. This indicates that the former may be in consideration, while the latter may not have been approved. The four "dropped" projects are for human resources development to improve secondary education, private sector development, natural resource management, and a Belize Social Protection Inclusion Project. One "pipeline" project is for COVID-19 vaccines, while three are related to climate change and agriculture. Of the "closed" loan projects, six were for physical infrastructure, five for natural marine and terrestrial resources, three for power, two for health, two for the Social Investment Fund (a statutory body which funds various projects), one for agriculture, and only one for education: primary education. These findings are especially significant because the strategy document between Belize and the World Bank entitled, "Belize: Systematic Country Diagnostic" recognized the importance of tertiary/higher education, and

stated that secondary education was a priority entry point to "improving education and skills" [54] (pp. 87–88) in the country, but there have been no approved loans for the sector since. Furthermore, a closer look at the "Belize: Systematic Country Diagnostic" strategy document [54,55] shows a team of economists and contributors from other countries. The "pipeline", "dropped", and "active" projects, as well as nine of the "closed" projects, do not appear on the public list of approved projects [88].

Different groups or entities can access funding from the Bank as "exterior clients" for various projects, but there have been no approved loans to Belize's government between 2015 and 2021. Figure 1 demonstrates the funding disbursements to Belize throughout the relationship. Disbursement for one project is still being made [88]. Although there was a sharp dive in funding around 2004–2011, which almost went down to nothing and formed a bell curve on the line graph, it is now rebounding. The lines of the graphs in Figures 1 and 2 are not similar; the funds were in decline during most of the news data collection period. There is no apparent similarity between flows of funds from the Bank to Belize and news reports in Belize about the Bank, except maybe after the COVID-19 pandemic started. This is because the government of Belize made repeated public pleas and promises for COVID-19 relief financing courtesy of the World Bank Group.

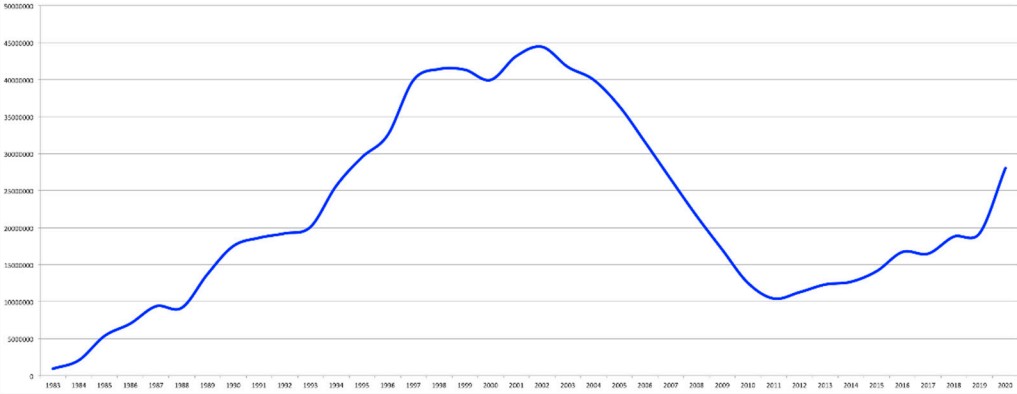

**Figure 1.** The World Bank (IBRD and IDA) Disbursements to Belize (1983–2020).

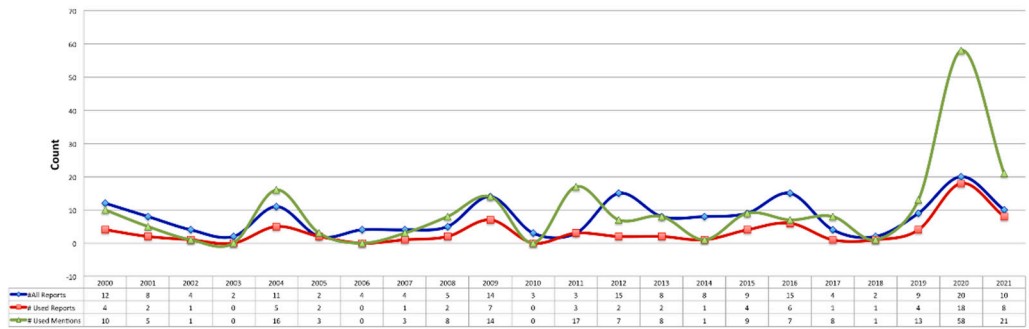

**Figure 2.** Distribution of "World Bank" Mentions (January 2000–October 2021).

### 3.2. Belize and the World Bank's Brand Relationship Form

The data collected from 2000 to 2021 in this research (Tables 2 and 3), reveal that the World Bank has been involved in all parts of the development operations of the Government of Belize: prime ministers, ministers, government departments, and statutory bodies. Aspiring politicians also use the name of the Bank in election promises of development funding [89,90] and even privately contract Bank personnel for development knowledge provision in campaigns [91].

**Table 1.** World Bank Projects in Belize

| | Project ID | Region | Country | Board Approval Date | Project Closing Date | Project Status | Project Name |
|---|---|---|---|---|---|---|---|
| 1 | P149522 | Latin America and Caribbean | Belize | 2016-09-12T00:00:00Z | 2022-05-31T00:00:00Z | Active | Energy Resilience for Climate Adaptation (GEF/SCCF) |
| 2 | P131408 | Latin America and Caribbean | Belize | 2015-03-03T00:00:00Z | 2020-09-30T00:00:00Z | Closed | BZ Marine Conservation and Climate Adaptation |
| 3 | P130474 | Latin America and Caribbean | Belize | 2014-09-29T00:00:00Z | 2019-09-30T00:00:00Z | Closed | Management and Protection of Key Biodiversity Areas in Belize |
| 4 | P127338 | Latin America and Caribbean | Belize | 2014-08-27T00:00:00Z | 2021-08-30T00:00:00Z | Closed | Climate Resilient Infrastructure |
| 5 | P132098 | Latin America and Caribbean | Belize | 2012-06-05T00:00:00Z | 2018-05-14T00:00:00Z | Closed | JSDF BZ Promoting Sustainable Natural Resource-based Livelihoods |
| 6 | P120349 | Latin America and Caribbean | Belize | 2011-03-01T00:00:00Z | 2016-01-15T00:00:00Z | Closed | BZ JSDF Improving Children's Health |
| 7 | P111928 | Latin America and Caribbean | Belize | 2010-09-16T00:00:00Z | 2016-11-30T00:00:00Z | Closed | Municipal Development |
| 8 | P117604 | Latin America and Caribbean | Belize | 2009-11-09T00:00:00Z | 2011-10-31T00:00:00Z | Closed | BZ Influenza Detection and Response (Standalone Trust Fund) |
| 9 | P103517 | Latin America and Caribbean | Belize | 2007-04-30T00:00:00Z | 2010-08-31T00:00:00Z | Closed | TFSCB Grant No. TF057206: Belize Statistical Development Project |
| 10 | P078216 | Latin America and Caribbean | Belize | 2002-10-16T00:00:00Z | | Closed | Community Managed Sarstoon Temash Conservation Project (COMSTEC) |
| 11 | P073924 | Latin America and Caribbean | Belize | 2001-04-03T00:00:00Z | | Closed | Social Investment Fund Project—Supplemental Loan |
| 12 | P040150 | Latin America and Caribbean | Belize | 2000-09-05T00:00:00Z | 2005-09-30T00:00:00Z | Closed | Roads and Municipal Drainage Project |
| 13 | P057045 | Latin America and Caribbean | Belize | 1999-03-22T00:00:00Z | | Closed | Northern Belize Biological Corridors (GEF-MSP) |
| 14 | P039292 | Latin America and Caribbean | Belize | 1997-03-18T00:00:00Z | 2003-03-31T00:00:00Z | Closed | Social Investment Fund |
| 15 | P006103 | Latin America and Caribbean | Belize | 1994-07-05T00:00:00Z | 1999-06-30T00:00:00Z | Closed | Power Sector Development Project (02) |
| 16 | P006104 | Latin America and Caribbean | Belize | 1993-11-30T00:00:00Z | 1997-12-31T00:00:00Z | Closed | Belize City Infrastructure Project |
| 17 | P006101 | Latin America and Caribbean | Belize | 1991-12-05T00:00:00Z | 1999-06-30T00:00:00Z | Closed | Primary Education Development Project |
| 18 | P006097 | Latin America and Caribbean | Belize | 1988-06-15T00:00:00Z | 1996-12-31T00:00:00Z | Closed | Agricultural Credit and Export Development Project |
| 19 | P006098 | Latin America and Caribbean | Belize | 1988-05-31T00:00:00Z | 1992-12-31T00:00:00Z | Closed | Road Maintenance & Rehabilitation Project (02) |

**Table 1.** *Cont.*

| | Project ID | Region | Country | Board Approval Date | Project Closing Date | Project Status | Project Name |
|---|---|---|---|---|---|---|---|
| 20 | P006094 | Latin America and Caribbean | Belize | 1986-08-05T00:00:00Z | 1993-12-31T00:00:00Z | Closed | Power Development Project |
| 21 | P006093 | Latin America and Caribbean | Belize | 1983-05-03T00:00:00Z | 1986-12-31T00:00:00Z | Closed | Road Maintenance and Rehabilitation Project |
| 22 | P152415 | Latin America and Caribbean | Belize | | | Pipeline | Belize FCPF REDD Readiness Preparation |
| 23 | P121004 | Latin America and Caribbean | Belize | | 2013-07-31T00:00:00Z | Pipeline | Helping the Q'eqchi (Keck-chee) Maya Thrive with Sustainable Forest Management |
| 24 | P177987 | Latin America and Caribbean | Belize | | | Pipeline | Belize COVID-19 Emergency Response Project |
| 25 | P172592 | Latin America and Caribbean | Belize | | | Pipeline | Climate Resilient and Sustainable Agriculture Project |
| 26 | P172956 | Latin America and Caribbean | Belize | | | Dropped | Belize Social Protection Inclusion Project |
| 27 | P006102 | Latin America and Caribbean | Belize | | | Dropped | NAT. RES. MGMT. |
| 28 | P006105 | Latin America and Caribbean | Belize | | | Dropped | Private Sector Development |
| 29 | P006106 | Latin America and Caribbean | Belize | | | Dropped | Human Resources Development Project |

Source: World Bank Group. [88]

**Table 2.** Frequency of Belize—World Bank Brand Relationship Indicators.

| Initiation | f | Emotion | f | Bond | f | Symmetry | f | Exclusivity | f |
|---|---|---|---|---|---|---|---|---|---|
| Involuntary | 129 | Joy | 31 | Grounded-Investment | 71 | Not Equal-WB Load | 37 | Not-Exclusive-other | 42 |
| Voluntary | 52 | Trust | 27 | Emotional-Dependency | 47 | Equal | 4 | Exclusive | 39 |
| Non-Voluntary | 29 | Anticipation | 24 | Grounded-Obligation | 18 | Not Equal-BZ Load | 1 | Not-Exclusive-IMF | 5 |
| | | Optimism | 17 | Grounded-Task | 17 | Not Equal-Unsure | 1 | | |
| | | Submission | 17 | Emotional-Superficial | 16 | Unsure | 1 | N/A | 4 |
| | | Disapproval | 13 | Emotional-Control | 14 | | | (blank) | 120 |
| | | Fear | 11 | Emotional-Domination | 11 | N/A | 42 | | |
| | | Sadness | 10 | Emotional-Intense | 5 | (blank) | 124 | | |

**Table 2.** *Cont.*

| Initiation | f | Emotion | f | Bond | f | Symmetry | f | Exclusivity | f |
|---|---|---|---|---|---|---|---|---|---|
| | | Acceptance | 9 | Emotional-Friendly | 3 | | | | |
| | | Admiration | 7 | Emotional-Additive Obsession | 2 | | | | |
| | | Anger | 7 | Emotional-Liking | 2 | | | | |
| | | Contempt | 5 | | | | | | |
| | | Grief | 5 | None | 4 | | | | |
| | | Annoyance | 4 | | | | | | |
| | | Love | 4 | | | | | | |
| | | Disgust | 3 | | | | | | |
| | | Ecstasy | 3 | | | | | | |
| | | Interest | 3 | | | | | | |
| | | Apprehension | 2 | | | | | | |
| | | Boredom | 2 | | | | | | |
| | | Remorse | 2 | | | | | | |
| | | Surprise | 2 | | | | | | |
| | | Aggressiveness | 1 | | | | | | |
| | | Vigilance | 1 | | | | | | |
| Total: | 210 | Total: | 210 | Total: | 210 | Total: | 210 | Total: | 210 |

Source: own research.

**Table 3.** Frequency of Belize—World Bank Brand Relationship Rewards.

| Government of Belize Relationship Rewards (2000–2021) | f |
|---|---|
| None | 62 |
| Instrumental–Functional Ties–Short-Term Goals | 25 |
| Both–Socio-Emotional and Instrumental | 24 |
| Instrumental–Functional Ties–Attain Long-Term Objectives | 24 |
| Socio-Emotional–Psychosocial Identity Functions–Reassurance of Self | 21 |
| Socio-Emotional–Psychosocial Identity Functions–Announcement of Image | 11 |
| Socio-Emotional–Rewards of Stimulation | 11 |
| Socio-Emotional–Psychosocial Identity Functions–Social Integration | 7 |
| Socio-Emotional–Rewards of Guidance | 6 |
| Socio-Emotional–Rewards of Social Support | 4 |
| Socio-Emotional–Rewards of Assistance | 3 |
| Socio-Emotional–Rewards of Security | 1 |
| Unsure | 1 |
| N/A | 6 |
| (blank) | 4 |
| Total: | 210 |

Source: own research.

The results, documented in Table 2, show that the main brand relationship form [53] between Belize and the World Bank can be called an "arranged marriage." This is based on the perspective of GOB personnel (Table A1) towards the Bank. In kin [53], the relationship is repeatedly inherited from the first administration, which was led by Prime Minister George Price of the People's United Party (PUP). This agrees with Fournier's [53] (p. 362) finding that the "arranged marriage" type is a "non-voluntary union." Fournier [53] (p. 362) adds that this relationship form was "intended for long-term, exclusive commitment," but has "low levels of affective attachment." All of these traits are found in this research. The results also reveal minor characteristics consistent with other relationship types, such as "dependency", "enmity", and "enslavement", which, Fournier says, "persists because of circumstances" [53] (p. 362). This is due to the experiences of Belize government personnel and the multiple roles they play in the banking cooperative. Membership in the World Bank Group probably started as a "marriage of convenience" to get development funding and support.

Tables 2 and 3 show the results of Fournier's [53] brand relationship indicators. Details of these results follow.

### 3.2.1. Initiation

Brand contacts, as evidenced in mentions (Table A2) of this arranged marriage relationship type, are mostly initiated involuntarily because of the government's need to meet the needs of Belize and Belizeans. In other words, the relationship is maintained "by preferences of third party" [53] (p. 362).

### 3.2.2. Emotions

The nature of the brand relationship type is mostly characterized by emotions of "joy," achieved after waiting in "anticipation" with "optimism" and "submission" for funding and recognition, but it is fueled by "trust" in the Bank. The follow quotation shows the complexity of the relationship.

> "We have been waiting for this project to get off the ground ever since 2009, when the Prime Minister negotiated the loan with the World Bank, and then the loan agreement was signed in December of 2010. And we have been waiting. It is now really excitement for us. We want to really see the works get underway now and we believe that within the next week or two, we should be seeing some works on the ground. On behalf of the mayors of Corozal Town, San Ignacio,

Santa Elena and Orange Walk Town, we want to express our gratitude. Most of us were waiting for this, as I said earlier, to be on the go for a long, long time. But anyway, it's here now and we want to say, thanks."

Simeon Lopez, Belmopan Mayor, Belize [92]

In another example from the qualitative data collected, Belize did not qualify for World Bank COVID-19 relief funds simply because of its WB classification in the International Bank for Reconstruction and Development (IBRD). Branded concessionary loans to help with the global pandemic were only earmarked for the poorest countries of the International Development Association (IDA), Belize's first membership category. PM Barrow was overjoyed and overtly thankful when he successfully negotiated the redirection of funds from the 2014 "P127338—Climate Resilient Infrastructure" project [93].

"Well, I had an extremely productive phone call with Madam Tahseen Sayed Khan, who is the World Bank's country director for the entire Caribbean, and I am very pleased to be able to announce, thanking her as I do so, that in consequence of that conversation twenty-one million U.S. dollars, that was part of the Climate Resilient Infrastructure Project program, twenty-one million US dollars will now be diverted . . . All that will now be able to go for assistance to be put into the pot [for COVID-19]."

Dean Barrow, Belize Prime Minister (2008–2020) [93]

The results show that emotions of "disapproval," "fear" and "sadness" are expressed when development needs are not met, either directly or indirectly because of the Bank's hesitancy, demonstrated lack of understanding, or use of inappropriate measuring instruments. The IDA and IBRD loans, and their requirements, are World Bank branded financial products and mechanisms that show a lack of thinking and understanding behind them. The World Bank Group bases loans on a country's gross domestic product (GDP) instead of their visible circumstances. These things cause failure to identify real needs and those countries, which should get access to grants, concessionary loans, and cheaper financing, especially in times of emergencies, remain in need. Belize believes that the vulnerability to disaster, climate change, and COVID-19 crisis is what truly shows their need, not their GDP [94].

"We might think Belize is a poor, developing country. We have a high debt; our income per capita is five thousand per annum. But, by international standards, we are considered middle income. We are considered a middle-income country, so we don't get eligible for concessionary funding from certain multilateral entities, so we don't benefit from cheaper financing. When these things hit us, it has a severe impact on us. And so because of that, we should be eligible for debt relief. And that is where the focus has been. You don't measure just by income, but you measure by the vulnerabilities."

Chris Coye, Finance Minister of State, Belize [94]

"Trust" in the Bank oftentimes leads to GOB's "submission." This point is evident in the examples of the government's awareness of Belize's repeated low rankings in the World Bank's "Doing Business" report [95,96], and their counteractive investments, policies, and actions to improve it. Changes have been made, such as, "stamp duty reform; exchange control reform; and lands reform, . . . modernizing the companies' registry and our companies' law" [97]. They also changed the law, based on World Bank's recommendations, to attract investors to Belize [96].

"In truth, as a nation, we are ranked one hundred and sixty-six out of a hundred and ninety countries in the World Bank's ease of doing business index for starting a business. I mean, we are really in bad shape at a hundred and sixty-six. But we need to fix it and that is exactly what we are attempting to do here today. For example, according to the World Bank, the average cost of starting a business is estimated around thirty-four percent of per capita income. Now, the cost in

Belize of forming a company to start your business would be as much as five hundred and eighty-four dollars. It is bad enough that it is high, but it isn't just that you go and pay five hundred and eighty-four dollars, but you have to go and make four different payments to be able to register your company. Making it easier to do business in Belize means that we have to make it simpler, cheaper and more efficient."

Juan "Johnny" Briceño, Belize Prime Minister (2020–present) [96]

Successive governments of Belize believe that what the World Bank and other multilateral development and financial institutions think of Belize, as evidenced by their global ranking systems or the level of confidence they have in the government, is very important. One government official even called the "Doing Business" report their "report card" [95]. Sadly, he further expressed his belief that they could not report on their own progress [95]. This shows the strong bond that Belize government personnel form with this WBG-branded financial knowledge product and the meanings they derive from it: the brand relationship [53]. Emotions of "fear" are sometimes expressed by the government because these World Bank branded products and content impact international perceptions of Belize—the country brand [98], and this is further tied to funding and foreign direct investment, which countries rely on to buy needed international goods. The World Trade Organization also uses the rankings of countries from the "Doing Business" report.

"We are a member of the IMF; Belize, is indeed. We are a world member of the World Bank, the IDB and so on . . . Because this is the way the world works. The truth of the matter is unless you have some kind of imprimatur from the IMF; at least an understanding of what you're doing, none of the rating agencies, none of the international financial institutions, will pay any attention to Belize. And we need to have that restoration of full confidence in Belize."

Said Musa, Belize Prime Minister (1998–2008) [99]

However, the World Bank Group has just discontinued its "Doing Business" report due to irregular data in the 2018 and 2020 publications [100]. Machen, Jones, Varghese, and Stark's [101] internal investigation found conflicts of interest and corruption by senior administrators of the Bank, who directly and indirectly acted to pressure the report's team and manipulate the methodology to improve the ranks of China and Saudi Arabia as favors, and demote Jordan and Azerbaijan. Azerbaijan's new placement was denied out of contempt that the country was actually making improvements. The ranking of the United Arab Emirates improved because of the subjective data fixing of other countries.

### 3.2.3. Bond

The repeat situation of waiting for funding and ranking improvements, described above, creates a lot of emotional dependency in government personnel towards the Bank, but while this research reveals that "emotional-dependency" is a major issue, "grounded-investment" is the bond that is largely responsible for sealing this arranged relationship. Additionally, "grounded-obligation," "grounded-task", and "emotional-superficial," which are high on the list, reveal little real emotional affect in the relationship. The bond is mostly based on funding, not real attachment. The bond for the "Doing Business" report is also tied to the possibility of investment and financial gain.

### 3.2.4. Symmetry

The symmetry between the GOB and the World Bank to fund development in Belize is "not equal" (Table 2). The Bank carries the initial load for meeting development funding needs for Belize and Belizeans. "Initial" because loans and other types of financing, except for grants, need to be repaid. This research reveals evidence that GOB is laden with the burden of repaying multilateral development loans, which have been as much as half of the government's debt obligations [102]. This causes its own financial crisis between the

government and lenders, such as the World Bank, which further affects their ability to meet development-funding obligations in the country.

When they cannot repay, the World Bank and the International Monetary Fund (IMF) have various solutions. The Bank has facilities to lend money to repay pre-existing loans, such as sovereign debt relief [103], but over borrowing can actually make the debt situation worse [94]. Another solution is financial restructuring and advice for that restructuring. Former Prime Minister Manuel Esquivel of the United Democratic Party (UDP) claims that with the advice of Belizean economists he instituted a Value Added Tax along with other austerity measures, and won against his own looming debt crisis [104]. However, Esquivel became known as the retrenchment prime minister because he fired close to eight hundred public officers in 1995 and he announced it right before Christmas [105]. Although PM Esquivel had his reasons, the move was especially egregious because this time is considered a merry season when families have many financial obligations.

Former UDP Minister Hubert Elrington, who served with PM Esquivel, explained the level of commitment that the Bank requires when it delivers advisory financial restructuring services. According to him [106], Bank personnel asked them to sell off utility companies to meet debt obligations; it is a tactic, which is based on neoliberalism development economics policies. The delivery of the World Bank message and the message itself, based on Elrington's "lived experience" [53], can be described as rude, exacting, and culturally offensive. It is not a partnership.

> "Let's say this country gets into a depression and we need help from the World Bank, we need help from the IMF; the first thing they are going to tell you when you get there, sell off the utilities ... When you're dealing with the IMF and when you're dealing with the World Bank, they come to the Cabinet and they get right in your face and they say look if you want help from us, you have to be committed. We don't want support; we want commitment from you. And you know what commitment is? Let us give you an example of what commitment is. Have you ever had ham and eggs? The chicken is supportive but the pig is committed. We want you to be like the pig...."
>
> Hubert Elrington, Former Housing Minister, Belize [106]

There is evidence that under the Musa Administration, the government allowed the World Bank, and the IMF, to "take over" in the midst of an economic debt crisis and allegations of corruption: the misuse of multilateral funds. At the announcement of a government audit and possible retrenchment, reporter, Janelle Chanona even asked Prime Minister Said Musa of the People's United Party (PUP), who was in charge, " ... are other players asking Belize to do this, the IMF, the World Bank?" [99]. PM Musa denied it, but later that year when the Opposition, United Democratic Party (UDP) brought up corruption claims, Musa asserted that he "invited" the World Bank and other international financial institutions to see the government's operations [107]. He denied following their advice.

> "They are coming because I invited them. And they are not coming to prescribe medicine for Belize. They are not coming to give us the usual IMF recipe for fiscal, structural adjustment, wage freeze, usual things, retrenchment, tax increases. No. The objective of this IMF mission is, first of all,—it is outside of the Article IV consultation; it is assisting us ... We invited all of them including the CDB [and World Bank] to come down and look at what we are doing; look at what we call the home grown fiscal stabilization and debt project that we are doing to manage our debt and also to bring down the fiscal deficit. We want them to do it."
>
> Said Musa, Belize Prime Minister (1998–2008) [107]

No one really believed this. The World Bank funding was apparently cut around the same time as these meetings with the Musa Administration, and they followed the advice of the IMF/World Bank to sell off assets and reduce staff. Residual flows of cash continued to flow from pre-existing loans (Figure 1), but this loss of confidence led to disqualification for new loans. In 2005, former PM Esquivel of the Opposition, had this to say about the

Musa Administration losing its financial credibility in the face of Belizeans, international financial institutions, and potential investors.

> " . . . The problem that the PUP [The People's United Party] government has is a lack of credibility. It's the lack of willpower to do what it necessary. I believe you [reporter, Stewart Krohn] yourself have said that on occasion? . . . Why they keep saying, we are rejecting the IMF, this is homegrown, when everybody can see that it is identical to what the IMF is doing. The added difficulty is they are trying to implement what the IMF agrees needs to be done, but at the same time they are pretending they can do it without the help of the IMF—and I'm talking about financial help. And they [PUP] themselves have said, without the imprimatur of the IMF, at this stage of the game, nobody is going to trust them, nobody is going to believe them. And so our position has always been, that is why we insist, there has to be a resignation of government."

> Manuel Esquivel, Former Belize Prime Minister (1984–1989 and 1993–1998) [104]

Due to the devastating impact of the retrenchment on citizens in 1995, PM Esquivel was voted out of government in the 1998 general elections and removed from party leadership. His pressure to retrench GOB staff in order to reduce government spending actually came from the International Monetary Fund (IMF) at the threat of devaluation.

As former PM Esquivel predicted [104], his party's ensuing Prime Minister, Dean Barrow of the United Democratic Party (UDP) government, regained financial credibility. PM Barrow repeatedly claimed ownership of re-connecting the funding ties with the World Bank [87,108,109], after successfully building up his own credibility following the Musa administration, but PM Barrow too faced the wrath, with the WBG and IMF telling his government to sell utilities and stop subventions, even in agriculture [110].

Abbakumova [74] adds that advice from the IMF and the WGB are not binding because these multilateral organizations are not supranational. This may be true but many believe that PM Esquivel was in an impossible situation under the pressure from the multilateral institutions, and he chose to save the country's economy instead of his political seat [105].

### 3.2.5. Exclusivity

In exclusivity (Table 2), this research finds that external funding options for projects available to Belize are not remarkably exclusive to the Bank; this can be a good thing. Based on "world bank" mentions for projects without any other international funding source or country: exclusivity, they have 39 or 43% of the mentions with 42 or 47% being "not exclusive—other." However, if the exclusivity results that also mention the International Monetary Fund (IMF) are added, then World Bank exclusivity rises to 49%. The sad part is that oftentimes, GOB funds one project in a piecemeal, wait and see, patchwork way, with funding from more than one organization, as it tries to navigate and appease the rules and waiting time of the World Bank and others [111].

On the bright side, these exclusivity results (Table 2) also mean that Belize can become less financially reliant on the Bank, while still remaining in the membership, which provides much needed or perceived, instrumental and socio-emotional rewards (Table 3).

### 3.2.6. Rewards

The relationship rewards part (Table 3) of this research also finds that 62 or 30% of the mentions about development funding offer "none" in terms of rewards, while "short-term goals" and "long-term objectives" are met at a much lower rate.

### 3.3. *World Bank Brand Relationship's Impact on UN Sustainable Development Goals in Belize*

The frequency of the 17 United Nations Sustainable Development Goals (SDG) referenced in the World Bank mentions (Table A2) from 2016 to 2021 are recorded in Table 4. These results do not reflect the actual impact that the Bank has made on them in Belize. A cross-comparison of relationship rewards of the brand relationship results in this research

and the SDGs reveals more about the fulfillment of these goals. Figures 3–5 show the cross-analysis results.

**Table 4.** Frequency of United Nations Sustainable Development Goals.

| United Nations Sustainable Development Goals (2016–2021) | f |
|---|---|
| GOAL 17: Partnerships to Achieve the Goal | 34 |
| GOAL 3: Good Health and Well-being | 19 |
| GOAL 13: Climate Action | 10 |
| GOAL 11: Sustainable Cities and Communities | 7 |
| GOAL 1: No Poverty | 7 |
| GOAL 9: Industry, Innovation and Infrastructure | 6 |
| GOAL 8: Decent Work and Economic Growth | 6 |
| GOAL 10: Reduced Inequality | 6 |
| GOAL 12: Responsible Consumption and Production | 6 |
| GOAL 2: Zero Hunger | 5 |
| GOAL 14: Life Below Water | 1 |
| GOAL 4: Quality Education | 1 |
| GOAL 5: Gender Equality | 0 |
| GOAL 6: Clean Water and Sanitation | 0 |
| GOAL 7: Affordable and Clean Energy | 0 |
| GOAL 15: Life on Land | 0 |
| GOAL 16: Peace and Justice Strong Institutions | 0 |
| Total: | 108 |

Source: own research.

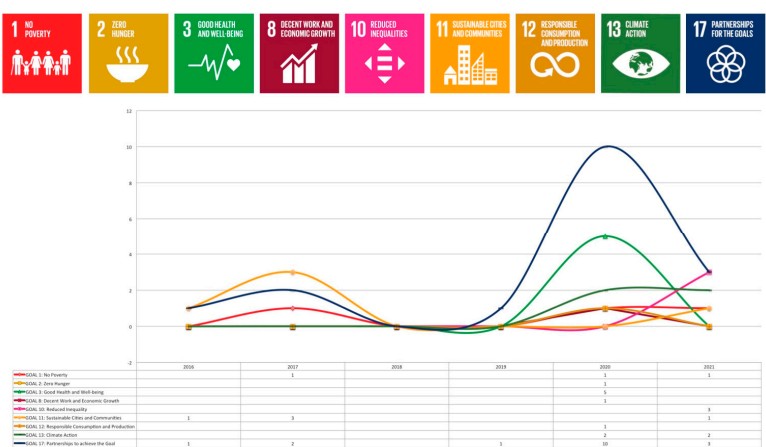

**Figure 3.** Nine of seventeen Sustainable Development Goals Not Achieved "None" Rewards (2016–2021).

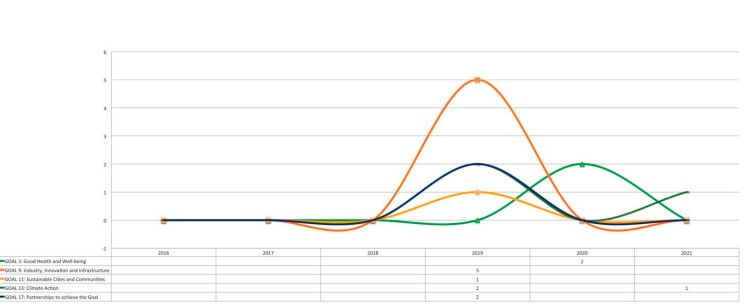

**Figure 4.** Five of seventeen Sustainable Development Goals Achieved "Long Term" Rewards (2016–2021).

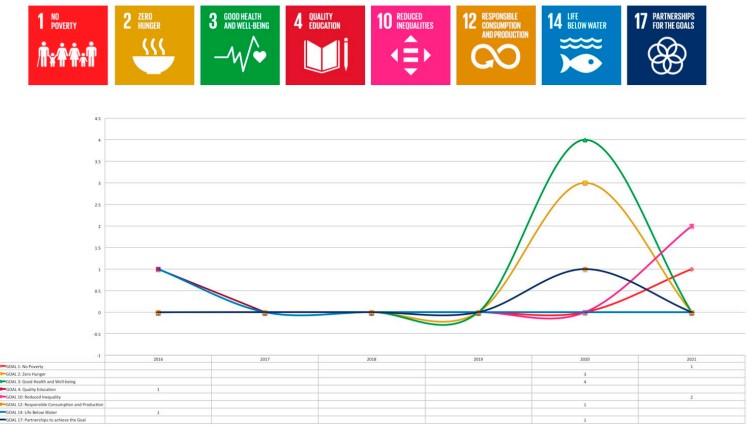

**Figure 5.** Eight of seventeen Sustainable Development Goals Achieved "Short Term" Rewards (2016–2021).

Nine of the seventeen SDGs are largely not achieved in this brand relationship; they register as "none" in terms of rewards (Figure 3). Only five of the seventeen SDGs register some "long-term" rewards (Figure 4), while just seven of them have some "short-term" rewards (Figure 5). The subjects of five goals were not recorded in this research. They are: Goal 5: Gender Equality; Goal 6: Clean Water and Sanitation; Goal 7: Affordable and Clean Energy; Goal 15: Life on Land; and Goal 16: Peace and Justice Strong Institutions.

The following excerpt from a News 5 news report on maintenance of the infrastructure of one highway and making it climate resistant provides a good example to illustrate the deeper problems in funding sustainable development. Reporter Mike Rudon asks, "Sir, why start on that section of road, which is not as bad as the area further up?" [111]. The Belize Ministry of Works' CEO's answer is alarming.

"As I mentioned, this project was conceptualized over five years ago, and this is the section that was being targeted. That other area you just mentioned will come under a different project and a different funding agency. We have the World Bank and the Government of Belize doing what is known as a climate resilience project that will then go from the airport junction to mile twenty-five. That should start later this year. We have seen the condition of the road deteriorating indeed. If you can remember just before the rains the Ministry of Works was out there doing a lot of spot patching—the area looks like a checkers board. Now that the rains have stopped we will be out there again. We will be doing some spot patching with the pre-mix."

Errol Gentle, CEO, Ministry of Works [111]

This answer further shows that the actual application of the WB-funded climate resiliency project may have no real rewards or positive impact on actually fulfilling its climate change protection, sustainable development goal. Any specific requirements to make the entire road safe in disaster conditions or "climate resilient" cannot possibly be enforced because the entire road is not being worked on at the same time. Additionally, with time spent waiting for more funding and depending on the quality of the first job, road conditions can also deteriorate in the different sections even before they get the WB funds and continue work. This research also reveals that some of those expected funds were diverted for COVID-19 relief because Belize did not qualify for IDA-branded COVID-19 relief funding.

Many of the nine SDGs, which measure with "none" in terms of rewards in this research, are critical to sustainable human development: Goal 1: No Poverty; Goal 2: Zero Hunger; Goal 3: Good Health and Wellbeing; Goal 8: Decent Work and Economic Growth; Goal 10: Reduce Inequalities; and Goal 11: Sustainable Cities and Communities. Goal 3: Good Health and Wellbeing, only shows up twice in "long-term" rewards. Goal 1:

No Poverty; Goal 2: Zero Hunger; Goal 3: Good Health and Wellbeing; Goal 4: Quality Education; and Goal 10: Reduce Inequalities are the sustainable human development goals which have achieved some "short-term rewards".

Goal 17: Partnership for the Goals, which is a must between the World Bank and any developing country, shows up 17 times for "none" in terms of rewards (Figure 3), but only two and one times for "long-term" (Figure 4) and "short-term" (Figure 5) rewards, respectively. Goal 1: No Poverty and Goal 4: Quality Education, which are related to the World Bank aims, each only show up once in "short-term" rewards (Figure 5) results and are missing in "long-term" rewards.

## 4. Discussion

Fournier [53] advances three main theories about brand relationships concerning their bonds, type/form, and quality. The use of her findings on brand relationship forms, detailed in question two of the results, determines that Government of Belize personnel perceive that the country has an "arranged marriage" [53] with the World Bank Group. The results on the characteristics of the relationship show that it has been maintained involuntarily by third party obligations, with bonds based on investments, not genuine positive feelings, which are needed in a true partnership. The Bank has the main investment obligation in the relationship, but it does not have complete exclusivity for development funders. Emotions of joy dominate the relationship, but this is curtailed by positive, negative, and coerced feelings of trust, anticipation, optimism, submission, and disapproval.

Fournier's last theory on the quality of the brand relationship helps to discuss and validate the findings in this research: brand relationship quality (BRQ) leads to relationship stability/durability [53] (pp. 363–365). She posits that reciprocal actions of consumers and brands lead to brand relationship quality, which is better than brand loyalty because it is holistic and offers conceptual richness with meanings. BRQ has six possible facets: "love and passion", "self-connection", "interdependence", "commitment", "intimacy", and "brand partner quality". "Brand partner quality" is more relevant to the "arranged marriage" brand relationship type because both are in the marital domain. Furthermore, Fournier [53] found that the presence or lack of consequences, such as "accommodation", "tolerance/forgiveness", "biased partner perceptions", "devaluation of alternatives", and "attribution bias" determine the stability or durability of a brand relationship.

A theoretical discussion on brand relationship quality (BRQ) [53] based on reciprocity, quality, and durability of the brand relationship between Belize and the World Bank found in this research follows.

### 4.1. Brand Relationship Quality Theory Analysis—Reciprocal Actions

The flows of money dropped significantly around 2004 when the Said Musa Administration could not pay its multilateral debt. Based on the reciprocity principle, found in all relationships [53], the World Bank expects that repayments of loans and associated fees must be made. Belize failed to do so and the World Bank cut access to new loans. There is little "tolerance" or "accommodation" exhibited by the Bank for lack of payment. They have restructuring alternatives in knowledge services and other loans, but the results show that these just make the situation worse and negatively impact citizens as well as the government's political future. In another alternative, the results document the terrible language used in the delivery of financial restructuring knowledge services and the total commitment that the World Bank requires of members to follow. Belize has always risked getting its dollar devaluated if the government does not follow World Bank advice or maintain good economic standing. These findings also discredit the touted partnership relationship that the World Bank uses in its branding. However, each future Government of Belize knows that if it improves its financial status, as another reciprocal action, it will be eligible to receive more funded projects as rewards. This is exactly what has happened, as each new Prime Minister is elected, but it is a dysfunctional cycle.

In another example, when GOB personnel thought that they would obtain improved ranking in the "Doing Business" report, after making improvements, based on World Bank knowledge recommendations, it did not really work out. Expected reciprocal actions by the World Bank Group did not happen as expected. Meanwhile, the multiple internal findings of corruption and data fixing in the 2018 and 2020 publications of the World Bank "Doing Business" report, which lead the Bank to suspend it, also show that the Bank's personnel are not living up to their reciprocal roles. They need to be reliable, dependable, and transparent to member countries and potential investors who rely on their credibility. While the Bank can cut funds when countries do not repay on time, what can countries do when the bank is found to be corrupt and subjective? Although it is good that the Bank pursued an independent, internal investigation, what is really at risk for the Bank? What immediate reciprocal actions can member countries take when their global country brand perceptions are at stake? World Bank Group reports, which are purported to be scientifically calibrated, objective, and credible, contain many country-branding messages. These branding messages become the foundation of many of these countries' brand perceptions. Machen, Jones, Varghese, and Stark [101] found that the indices, ranking, and even the methodologies were vulnerable to influence, bribery, subjective will, and manipulation. Moreover, the Bank had no internal checks and balances [101]. Meanwhile, countries without influence allow their branding to be defined by the Bank. But though the Bank vows to create a replacement in its bid to cut the negative brand association, these high level, internal, corruption findings, which also implicated a past president, should erode some of the trust and importance that the government of Belize, other countries, "stakeholders," "policymakers," the "private sector, civil society, academia, journalists, and others" [100], place on the Bank's administrators for credible knowledge products and services. Undoubtedly, this leaves a credibility gap, which is similar to Teferra's [18] claim about higher education concerning what parts of the previous ranking lists or knowledge products from the Bank can still be believed. There is a lot of tolerance and forgiveness by members towards the Bank. In any case, it is unfair that while some countries were influencing senior Bank officials to rank high, and some Bank officials were using personal agendas to demote countries, Belize's government was following the rules and doing its best to place better in order to attract investment and new industries.

*4.2. Brand Relationship Quality Theory Analysis—Brand Partner Quality*

This brand relationship research reveals very little affect toward the Bank as a brand; therefore, there is little chance of any of the components of "brand partner quality" that Fournier describes.

> "The notion of brand partner quality is suggested here as an analogue, one reflecting the consumer's evaluation of the brand's performance in its partnership role. The strong-brand stories suggest five central components of brand partner quality: (1) a felt positive orientation of the brand toward the consumer (e.g., making consumer feel wanted, respected, listened to, and cared for); (2) judgments of the brand's overall dependability, reliability, and predictability in executing its partnership role; (3) judgments of the brand's adherence to the various "rules" composing the implicit relationship contract (Sabatelli and Pearce 1986; Wiseman 1986); (4) trust or faith that the brand will deliver what is desired versus that which is feared; and (5) comfort in the brand's accountability for its actions."
>
> [53] (p. 365)

There are a lot of issues revealed in this research about the Bank's "felt positive orientation", "dependability", and "adherence" to rules. The brand relationship shows that it is bonded by "grounded investment," but although there are emotions of "joy" and "trust", "anticipation", caused by waiting for the Bank, is also a leading orientation. The government has a lot of trust in the Bank, but the Bank does not always give back desired outcomes. When Belize government personnel cannot perform their development duties on

Belize to meet its obligation, due to waiting for the Bank to use the right methods or product branding to measure and determine funding needs, there is also a breach of partnership rules. Based on the Bank's branding, these rules concern ending extreme poverty and being a development partner, but the results show that even waiting for disbursement of loan funds, after approval, takes too long for the country to realize the real benefits of the loans or sustainability aims.

The Bank's work as a funder for the United Nations' Sustainable Development Goals makes its partnership role in these funding relationships even more crucial. The UN places a lot of emphasis on meeting development needs, but GOB's perception of the Bank about meeting the SDG, is also very poor. The qualitative results of the rewards data show, that even when funds are being distributed for sustainable development projects, the amount of funds and its application does not mean that it is actually meeting the desired goals.

*4.3. Brand Relationship Quality Theory Analysis—The Stability/Durability*

This research reveals that the World Bank Group has penetrated every facet of the Government of Belize's development agenda with mostly open access. The Bank has asked Belize's government to sell off its public utilities, fire public service staff, stop subventions, and take other measures to save money in financial crises. These findings provide evidence of the first-generation and the second-generation of development economics policies [20] in Belize, which is also similar to actions in other developing countries. Additionally, Belizean political aspirants use the name of the Bank to win elections. Once elected, the "world bank" is used in budget speeches, to rescue the country from natural and financial emergencies, along with further promises for development. It is also used as a powerful imperative to justify funding and the change of laws. The two major political parties (UDP and PUP) also fight between themselves while using the name of the Bank to improve their perceived credibility against the other in the eyes of Belizeans, investors, and the Bank.

However, the Bank has failed to truly develop its touted partnership relationship with Belize to alleviate poverty and build knowledge with sustainable human development. The World Bank funds mostly physical infrastructure and natural resources in the country.

The Government of Belize seems to tolerate a lot from the World Bank while remaining hopeful that things will change. The results suggest that the Bank has more rules for countries as members and clients than these countries have for the Bank's personnel. The lack of equal voting power by all members and its tie to money, shows that certain countries will always control decision-making processes and their outcomes. For example, in the World Bank Group institution, called the International Bank for Reconstruction and Development (IRBD), Belize needs 313.2 times its 0.05% voting power to reach the 15.66% voting power of the United States. The needs and recommended changes to the multilateral system proposed by Belize and other countries with minute voting power may never be met. If all countries had one vote of equal weight it would help to make the Bank more accountable. It would also help to make the membership feel more like a partnership cooperative. It would also help the Bank fulfill its promise of shared prosperity.

Although GOB is also a member of the cooperative, which has its own requirements and is not affected by funding, the main purpose of the bond between the officials and the Bank is to obtain funding for the development of Belize and Belizeans. In this example, the durability of the entire relationship does not depend on funding, but, paradoxically, funding is the whole purpose. Belize has funding alternatives, but the Bank is still the largest development donor agency. This fact threatens to devalue the alternatives, but more attractive alternatives can threaten the durability of this WBG relationship [53]. Even when funding or access to loans is cut, the membership remains. Belize is still obligated to attend meetings, pay fees, and be loyal to WBG/IMF financial assessments. However, the country's loyalty to meetings, etc., does not mean that the brand relationship is good. As mentioned before, brand quality is better than loyalty [53] to ensure the strength and longevity of the relationship.

Rewards, as it pertains to ensuring long-term relationships, may not be important, because in Belize, with or without rewards, a divorce from the World Bank may require a legal separation and approval by Belizeans. This is because the country's membership is enshrined in the Laws of Belize: "International Financial Organizations Act, Chapter 265." Other country clients/members of the cooperative can also vote Belize out, but even after the Bank cut funding to the country during the Said Musa Administration, this did not happen. This membership adds an "entrapment" quality to the overall financial relationship. It is very hard to leave. Feelings of unhappiness about inadequate funding or lack of funding undoubtedly seeps over to the membership relationship, but as mentioned in these membership meetings, Belize has minute and ineffective voting power. This undoubtedly further negatively impacts the brand relationship.

Furthermore, Fournier created a figure with six panels showing alternative brand relationship trajectories [53] (p. 364). The line graphs in the panels are based on time and (perceived) closeness. She calls the bell curve in panel one, a biological life cycle depicting long-term brand relationships: a natural trajectory, where it begins, grows, reaches its plateau, and ends. Her panel one looks similar to the Belize–World Bank funding relationship (Figure 1), which is based on time and money disbursed. The resurgence of the funding relationship, as noted in the line graph (Figure 1), is due to adding value [53]. These new flows of funds represent investments in the relationship.

The evidence of low emotional affect, which would add quality to the brand relationship, suggests that there has been little perceived real brand building by the World Bank to Belize. While Belize's membership in the banking group remains with each change of the governments of Belize, similarly to panel six (stable maturity), but in a looping membership life cycle, the actual branding relationship may look more like a combination of panel two (growth–decline–plateau) and panel five (cyclical resurgence) depending on the individual experience of each government with the Bank. For one government, it may have even been a "passing fling", such as panel three. It is not impossible that as Belize's membership in the World Bank continues on a dysfunctional path, the funding relationship could look more like "approach—avoidance" or "cyclical resurgence" [53] (p. 364).

As to the durability/stability of this relationship, it will go on until one government has the courage to take the steps necessary to leave the Bank or at least stop borrowing from it. Depending on the Prime Minister, the government's policies, and the members of government, however, these multilateral development policies do not have to be followed. Belize also has other choices for development funding. Development economics frameworks do not all have to be multilateral; they can also be bilateral, homegrown, or mixed. The current Briceño Administration of the People's United Party has explained to the IMF that they are taking a homegrown approach to Belize's economic recovery [34].

## 5. Conclusions

Based on the extensive review of the literature and the results of this original relationship research, the researchers offer the following concluding remarks. In this multilateral financial relationship, the World Bank Group has been strategically controlling the way these so-called developing countries see themselves, and their status. This is what Fournier [53] (p. 367) calls brand relationships; it is "what consumers do with brands to add meaning in their lives." The Bank positions itself as their best solution to development funding and knowledge in its branding messages. The low interest rates are attractive. However, while it has been uplifting itself in the minds of these countries, it has also been failing to live up to its branding messages with positive reciprocal actions, which would lead to stronger and more positive brand relationships. The World Bank-branded loan products, as well as financial knowledge products and services in reports, indices, staff, and experiences, have been directly and indirectly controlling the meanings formed in the minds of these government personnel about themselves. Through this unsustainable relationship with the Bank, these debtor countries see themselves as needy, unable to develop themselves, and in need of funding and knowledge from the Bank. While these

leaders may seek to officially uplift the perception of the Bank, they are also bringing down their meaning or value in themselves, thus reinforcing the unhealthy brand relationship. The dysfunctional brand relationship does not make them feel like they are in control of their own development, but an unhealthy brand relationship from the perspective of these member countries may be a healthy relationship from the perspective of the World Bank Group because it helps to validate their existence. This type of relationship and the need created is clear and understood by both parties in capital markets where the transaction of goods and services for money is commonplace. But as the World Bank Group indiscriminatingly straddles the line between capital and social agendas, the implications are dire for those who sign into the relationship based on the promise of global social good. The banking group has also been controlling the potential of these countries to achieve real development, especially sustainable human capital development. As Bresser-Pereira [24] explains, the Bank is not acting on its own free will to fulfill its publicly stated development mission. It is also funding short-term projects [24] instead of long-term solutions, which Inikori [70] says is ideal. Majority shareholders of creditor member countries, who also have the highest voting power and dedicated executive directors and governors in the Bank, directly and indirectly, control its operations. Meanwhile, the majority of member countries are denied real impact. However, the dramatic changes in the world and access to information, as well as better-educated leaders, have forged a loss of confidence in the multilateral system which surrounds or controls the World Bank Group, as well as the negative reciprocal actions of the Bank itself. Many progressive leaders of these countries know what has been happening and they want change.

In the case in this research, the World Bank Group is failing to meet the sustainable human capital development needs of Belizeans in two ways: the actual funding relationship and the brand relationship. The brand relationship quality discussions reveal that the reciprocal actions create an unsustainable and dysfunctional brand partner relationship, which is biased, unfair, and debilitating to Belize, Belizeans, and the government. However, leaving the membership has many complex legal, financial, and brand difficulties.

### 5.1. Belize and the World Bank's Funding Relationship

This research makes it clear that although the World Bank Group promotes a global knowledge economy and human capital development, and is considered the "premiere" multilateral lender to Belize, these funds are not being used to improve higher education in the country. This is essential to building knowledge economies. The Bank funded one primary education project, but a sole secondary education proposal was dropped. These findings fall in line with evidence from other developing countries in Africa, and Asia [10,14,18], where historically, the Bank saw higher education as a luxury and only funded primary education. Meanwhile, although the population in Belize is thin as Carneiro's [54,55] World Bank strategy document on Belize describes it, and may not align with the World Bank Group's methodology, which looks at resulting financial gains, it is increasing. Informal evidence from Taiwan's scholarship programs in Belize, for example, shows that there is growing demand for higher education by Belizeans. The HE sector must also provide more programs.

### 5.2. Belize and the World Bank's Brand Relationship

The brand relationship findings conclude that the Government of Belize perceives that it has an unhealthy "arranged marriage" relationship with the World Bank Group. The Government of Belize would feel hindered or have great pains to ask the World Bank for funding for what it needs due to all the transparent and hidden rules, waiting, and a lack of genuine brand bonds involved in the relationship. There are no strong feelings of love or genuine partnership to keep this relationship going other than the disbursement of money. The overall revolving membership life cycle of offers, promises, knowledge services, and meetings creates broader aspects of "entrapment".

*5.3. World Bank Brand Relationship's Impact on UN Sustainable Development Goals in Belize*

The real issue concerning the strength of the relationship is how this multilateral brand relationship impacts sustainability: the sustainability of the funding relationship to achieve sustainable development funding goals, and the branding relationship to develop and maintain genuine brand bonds. The relationship rewards data of the brand relationship also emphasize that there is a lack of focus on sustainable development goals, such as education, work, partnership, health, and poverty, which together identify the poor strength of the brand relationship. SDG 17: Partnership for the Goals, should be at the heart of accomplishing funding for sustainable development. Teamwork is also a value of the World Bank Group, but this research suggests that actual partnership is missing. The Bank has Belize's loyalty due to legal aspects and historical ties in the membership, but it is not a quality brand relationship.

*5.4. Implications for Practice*

5.4.1. World Bank Group

Solving poverty by true partnerships for real shared prosperity in development require the building of genuine brand bonds. Building brand bonds must start with understanding the real history between the so-called developing and developed countries: what made the development disparity, the roles all countries played in it, and what continues to create poverty around the world. The reality of this debilitating and inhumane history needs its own brand of helpful financial concessions to countries in need of sustainable development. The World Bank Group must dismantle the historical rules that continue to reinforce the dominance of advanced economies on otherwise well-meaning, multilateral relationships. Personnel of the World Bank Group need to continue to advance and innovate their development economics policies and methodologies with qualitative methods, such as brand relationships so that they can work closer towards fulfilling the needs of Belize and other countries. Although the World Bank Group is dominant in the multilateral marketplace, managing its brand is important. To avoid irrelevancy, they must pay attention to what all member countries are saying about them and how they perceive the membership. Positive reciprocal actions must also be consistent so that the World Bank's branding messages and brand perceptions become more closely aligned. Branding initiatives by brand managers can help to shape the brand [53], but cooperatives, companies, countries, people, etc., that branding represent, must live up to them. As Machen, Jones, Varghese, and Stark [101] recommended, the banking cooperative also needs robust, internal mechanisms to safeguard its brand from presidents, directors, managers, and staff who do not share its mission and vision, and those who have personal, subjective, or biased intentions. Producing a new business and investment publication will not help the Bank's brand if it is still vulnerable to corruption. Member countries must also be given reachable avenues for advancement to achieve actual decision-making status in the Bank. Why must the appointment of a WBG president always come from the United States? If the Bank is indeed more involved in the business of creditor countries than debtor nations, as suggested in the findings of this research, or if developing countries are products in a process of manufacturing based on what creditor nations want, then the entire institution must be dismantled and redesigned by current members from the bottom up. All members must also be equal with identical voting power. The multilateral relationships among countries and the World Bank Group must be for real sustainable development to preserve our planet and people: it must not be for competition and capital gain.

5.4.2. Countries with Development Needs

Although certain members of the United Nations made repeated cries for change and improvement of the multilateral system, all members must do so. Under the present system, this is the only way that their recommended changes have a chance of becoming reality. These countries must also seek out and create other sources of funding so that reliance on the Bank is diminished or minimized. The results suggest that the World

Bank branded knowledge and funds may be another form of colonialism of the mind and potential. Member countries need to stop following rules that they did not make themselves, understand that they are not benefitting from them, and recognize their power in numbers. They must also demand equal voting power and collectively take the necessary sacrifices in actions to ensure it. It is better to band together in a branding that you give yourselves and stop repeating the word, "developing", because, in fact, all countries are developing. There is no one path, definition, or example of what it means to be developed. There is no one frontier of development. Countries must put citizens first, and invest time and effort into creating their own knowledge resources, models, and strategies for development. The scholars and leaders of these countries know more about their own situation than the handed down, filtered, pre-approved, and possibly corrupted knowledge from the World Bank Group and other multilateral institutions.

*5.5. Research Limitations and Recommendations for Future Study*

The data used in this brand relationship research are limited to found interviews and speeches of Government of Belize (GOB) personnel about the World Bank Group in news stories from one television station. It is based solely on their comments about the World Bank Group, its products, services, polices, meetings, guidelines, and people. This perspective was essential to study their brand perceptions of the Bank. The data collected are from January 2000 to October 2021. They do not include the initial membership period from 1982 to 1999. The researchers could not separate the different roles that the government has in the Bank to understand them separately.

The main limitation of this research means that the results come from one perspective: the government of Belize personnel about the World Bank Group. However, the raw data revealed a lot more about the networked relationships between/among member countries themselves, and with the World Bank Group in the multilateral cooperative. The Bank also has official and unofficial brand perceptions of member countries. Those corrupted Bank administrators who changed data in the "Doing Business" report were showing their true brand perceptions of the countries involved. They had the brand perception that some countries did not deserve a higher rank even though their unpublished scores proved that they improved, but other countries were perceived as more deserving even without the evidence. How much of Malpass's comments about risky and zero risk loans to countries come from facts, and how much comes from brand perceptions about the different categorizations of countries? There is an inherent conflict of interest in the dual roles of defining risks and providing loans, especially in a bank found to have no internal credibility mechanisms. Citizens and researchers of countries also have brand relationships with the World Bank Group and perceptions of the relationship between the Bank and their governments.

Further research is needed to look into different perspectives of the entire cooperative membership relationship as a solution to the main limitation in this research. This should add a lot to the scarce literature on development economics and what may be called membership brands or cooperative brands in the multilateral marketplace. Interviews could be conducted with members of present and past government administrations who are still alive and able to recall what transpired within the World Bank Group. Additionally, research is also needed which documents the World Bank's projects and services in Belize, especially in relation to how individual Prime Ministers of Belize and their administrations handled it. The work of the International Monetary Fund, which works as a big sister to the World Bank Group, and other multilateral organizations in Belize, also need to be revealed.

*5.6. Recommendations for Theory*

Fournier's [53] complex, but thorough, relationship theories, based on a two-party consumer brand dyad, were used as guides in this research because of the unavailability of a multilateral membership relationship theory. Based on the findings in this research, scholars must understand that the relationship circumstances of countries in multilateral

cooperatives are more complex and multifaceted than a dyad. Each government is a member, a shareholder, a client, a customer, and a consumer, and it is all happening at the same time. The Bank's financial products, services, content, global indices, people, and experiences form a complex network of bonds and associated brand relationships with country clients: their governments, their citizens, and researchers. Workers of the cooperative are also complex, with people who directly personify and indirectly manage the brand to member/clients, and they can also corrupt it. The researchers conclude that due to the various roles, there are multiple and different brand relationship trajectories that member countries experience at the same time while in the World Bank Group. These can be positive or negative, or both. They also impact and influence each other. As previously noted, more research is needed in this complex area, which would deliver a valuable framework and conclusive theory for researchers to use.

Brand analysis with qualitative findings adds new insights to development economics relationship research with more avenues to measure, analyze, and understand multilateral development economics funding problems. This is more beneficial to development economics policies than the usual quantitative financial study. Having funds does not mean that actual development needs and goals are being met. Moreover, the qualitative findings have verified the quantitative data. They have also given more clarity and meaning to the complex issues in the membership of the banking cooperative for more academic discussion. Although this research referred to citizens in need of sustainable human development, the results reveal that government personnel also need human development. They need strength and informed actions to challenge, disrupt, and change the multilateral system that was arranged for them and passed down to them; they did not design it. They need creativity, space, and time for research and experimentation to determine homegrown solutions that work. They need self-confidence to lead development of their countries with determination, vision, and resiliency in this changing world of opportunities for economic development in knowledge.

**Author Contributions:** Conceptualization, J.D.I.; methodology, J.D.I.; validation, J.D.I. and C.-H.Y.; formal analysis, J.D.I.; data curation, J.D.I.; writing—original draft preparation, J.D.I.; writing—review and editing, J.D.I. and C.-H.Y.; visualization, J.D.I.; and supervision, C.-H.Y. All authors have read and agreed to the published version of the manuscript.

**Funding:** This research received no external funding.

**Institutional Review Board Statement:** Not applicable.

**Informed Consent Statement:** Not applicable.

**Data Availability Statement:** Financial disbursement data and development project data used in this research are available on The World Bank Group's website at www.worldbank.org (accessed on 25 September 2022).

**Conflicts of Interest:** The authors declare no conflict of interest.

## Appendix A

**Table A1.** Profile of Government of Belize Personnel Data Samples.

| Position Titles | f |
|---|---|
| Prime Minister–UDP | 69 |
| Prime Minister–PUP | 29 |
| Ministers–PUP | 23 |
| Government Department Reps.–UDP | 16 |
| Former Ministers–UDP | 12 |
| Chief Executive Officers–UDP | 10 |
| Ministers–UDP | 9 |
| Mayors–UDP | 7 |
| State Owned Enterprises | 6 |

**Table A1.** *Cont.*

| Position Titles | f |
|---|---|
| Government Aspirants–PUP | 5 |
| Leader of the Opposition–PUP | 5 |
| The Opposition–PUP | 4 |
| Leader of the Opposition–UDP | 3 |
| Senators–UDP | 3 |
| Chief Executive Officers–PUP | 2 |
| Councillors/Mayor Aspirants–UDP | 2 |
| Former Prime Ministers–PUP | 2 |
| Former Prime Ministers–UDP | 2 |
| Government Rep.–PUP | 1 |
| Total: | 210 |

Source: own research.

## Appendix B

**Table A2.** Profile of Descriptions of "World Bank" Mentions Data Samples.

| Descriptions | f | | f |
|---|---|---|---|
| WB-BZ Confidence | 14 | WB-Loans-Approved-Waiting | 7 |
| WB-Call-Approved-Waiting | 6 | WB-Loans-Repayment | 1 |
| WB-Call-Seeking | 1 | WB-Meeting-BELIZE | 2 |
| WB-Campaign Promise | 7 | WB-Meeting-ONLINE | 1 |
| WB-Complaint | 19 | WB-Meeting-USA | 1 |
| WB-Conference/Meeting/Call With Investors-FOREIGN | 1 | WB-Money-Approved-Coming | 2 |
| WB-Document | 1 | WB-Money-Approved-Disbursed | 1 |
| WB-Funding-Complaint | 3 | WB-Money-Approved-Waiting | 1 |
| WB-Funding-Disbursed | 2 | WB-Money-Promised-Waiting | 3 |
| WB-Funding-Diverted | 2 | WB-Loans-Application-Waiting | 3 |
| WB-Funding-Not Approved | 2 | WB-Partnership | 4 |
| WB-Funding-Promised | 1 | WB-Policy | 10 |
| WB-Funding-Promised-Waiting | 5 | WB-Project | 7 |
| WB-Funding-Seeking | 8 | WB-Reports | 6 |
| WB-Funding-Talking | 9 | WB-Seeking Info For WB | 1 |
| WB-Global Indexes | 10 | WB-Service-Advice | 1 |
| WB-Grants | 3 | WB-Service-Budgeting | 1 |
| WB-Grants-Approved | 3 | WB-Service-Does Not Do | 3 |
| WB-Loans | 17 | WB-Service-Financial Restructuring | 9 |
| WB-Loans-Application | 2 | WB-Service-Spending Management | 1 |
| WB-Loans-Application-Waiting | 3 | WB-Sponsorship | 1 |
| WB-Loans-Approved | 1 | WB-Standards | 5 |
| WB-Loans-Approved-Coming | 2 | WB-WB/BZ-Loans Cut/Re-Established | 15 |
| WB-Loans-Approved-Disbursed | 5 | | |
| Total: | | | 210 |

Source: own research.

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
