# Peer review of "Arranged Marriages in Multilateral Partnerships—Investigating Sustainable Human Development Financing of Belize in the World Bank Group: A Brand Relationship Theory Approach"

_2199-8531, doi:10.3390/joitmc8040197_

Round 1
Reviewer 1 Report (New Reviewer)
According to the studies done, the following things to improve the quality of the manuscript. is presented:
1- The exact evidence of the method was not found in the abstract. Please improve.
2- In the required method section, specify why you chose this method (combined approach).
3- Compare the results presented in the conclusion section with the results of more research (consistent or different).
4- According to the limitations of such studies, please describe the limitation in detail and suggest applicable solutions for future research.
Author Response
Dear Sir or Madam:
On behalf of my co-writer, I would like to thank you for reviewing our first manuscript submission entitled: “Investigating Sustainable Human Development Funding of Belize in the World Bank Group: A Brand Relationship Theory Approach.” You gave valuable feedback, which has truly benefitted this extensively revised version.
After a request for a name change it is now tentatively titled, “Arranged Marriages in Multilateral Partnerships - Investigating Sustainable Human Development Financing of Belize in the World Bank Group: A Brand Relationship Theory Approach.” Please find our feedback to your comments and suggestions below.
Reviewer #1
- Abstract: The exact evidence of the method was not found in the abstract. Please improve
- Methodology: In the required method section, specify why you chose this method (combined approach).
- Conclusion: Compare the results presented in the conclusion section with the results of more research (consistent or different).
- Conclusion: According to the limitations of such studies, please describe the limitation in detail and suggest applicable solutions for future research.
Feedback for Reviewer #1
- Abstract: The abstract has been revised and updated with more information about the methodology.
- Methodology: The methodology has been revised. It now includes more detail as well as a reference in development economics to support why a combined quantitative and qualitative approach was chosen.
- Conclusion: Information was added to the conclusion about getting more information from interviews as a solution to the limitations of found information. Interviews would give more insights into what these government personnel experienced within the World Bank Group. This research also only offers one perspective. There are other perspectives within the multilateral membership, which need to be investigated such as that of the World Bank Group.
- Conclusion: The conclusion has been revised with limitations and a label to match. It now includes solutions to limitations encountered during the research.
Thanks again
Reviewer 2 Report (New Reviewer)
1. The title does not describe the outputs or outcomes of the research carried out, but only the approach taken is visible
2. In the abstract it is necessary to add a short narrative that explains the problems in this research
3. The introduction section needs to be equipped with the context of the object of research and a more detailed explanation of the SGDs that are used as a reference in the series of this research process
4. How can the branding carried out in this research be an effort for the World Bank related to the concept of Sustainable Human Development?
5. The data processing and methods used have not been explained in detail and seem too short to get the desired output in this study
6. Each table and figure still require an explanation of its relationship and also an explanation of its functions to the branding process carried out
Author Response
Dear Sir or Madam:
On behalf of my co-writer, I would like to thank you for reviewing our first manuscript submission entitled: “Investigating Sustainable Human Development Funding of Belize in the World Bank Group: A Brand Relationship Theory Approach.” You gave valuable feedback, which has truly benefitted this extensively revised version.
After a request for a name change it is now tentatively titled, “Arranged Marriages in Multilateral Partnerships - Investigating Sustainable Human Development Financing of Belize in the World Bank Group: A Brand Relationship Theory Approach.” Please find our feedback to your comments and suggestions below.
Reviewer #2
- Title: The title does not describe the outputs or outcomes of the research carried out, but only the approach taken is visible
- Abstract: In the abstract it is necessary to add a short narrative that explains the problems in this research
- Introduction: The introduction section needs to be equipped with the context of the object of research and a more detailed explanation of the SGDs that are used as a reference in the series of this research process
- Conclusion: How can the branding carried out in this research be an effort for the World Bank related to the concept of Sustainable Human Development?
- Methodology: The data processing and methods used have not been explained in detail and seem too short to get the desired output in this study
- Results: Each table and figure still require an explanation of its relationship and also an explanation of its functions to the branding process carried out
Feedback for Reviewer #2
- Title: Thanks for your request. We have updated the title to reflect your feedback. Please let us know if it is what you had in mind. It would be great if you could recommend a title for us to consider. However, we feel that a shorter title would be preferable. Previous title: “Investigating Sustainable Human Development
Funding of Belize in the World Bank Group: A Brand Relationship Theory
Approach”. Updated title: “Arranged Marriages in Multilateral Partnerships - Investigating Sustainable Human Development Financing of Belize in the World Bank Group: A Brand Relationship Theory Approach.”
- Abstract: The abstract has been revised and updated with more information about the problem of the research in general development economics as well as in Belize.
- Introduction: We have updated the introduction with more information about the knowledge economy. We believe that details about the 17 SDGs are actually outside the scope of this research because the focus is on challenges to fund them. We are mainly concerned with the human development goals, especially education. The human development goals are merely means to an end, and not the true objective of this research. The true objective of this research is knowledge economy, which must be achieved through sustainable human development.
- Conclusion: We believe that the World Bank Group must align its branding messages with its brand. They can do this by building genuine brand bonds with members. It starts with fulfilling its development and partnership promises, which includes sustainable human development. We have revised the section on practical implications in the conclusion, which offers more information.
- Methodology: The methodology has been revised. It now includes more details. On the contrary to what you commented, the method of benchmarking the brand relationship theories for analysis and using them as tools to measure the data surveyed have provided a lot of information in the results, as well as ideas for discussion and further research into the area of brands in multilateral relationships of development economics. The relationship metaphor is an excellent avenue to understand brands because we all build relationships with organizations, companies, products, and services that we use or desire not to use because of negative experiences.
- Results: The tables and figures in the results are properly labeled and referenced in the text. In part, the research aimed to understand the World Bank’s brand from the perspective of a member country. The Bank’s brand is not what it says about itself; that is its branding. The brand is what others, such as members, say about the Bank.
Thanks again
Reviewer 3 Report (New Reviewer)
1- author should add motivation and contribution in the introduction
2- author should highlight the study limitation
3- grammars and proofreading need improve
4-some of the references need to update
Author Response
Dear Sir or Madam:
On behalf of my co-writer, I would like to thank you for reviewing our first manuscript submission entitled: “Investigating Sustainable Human Development Funding of Belize in the World Bank Group: A Brand Relationship Theory Approach.” You gave valuable feedback, which has truly benefitted this extensively revised version.
After a request for a name change it is now tentatively titled, “Arranged Marriages in Multilateral Partnerships - Investigating Sustainable Human Development Financing of Belize in the World Bank Group: A Brand Relationship Theory Approach.” Please find our feedback to your comments and suggestions below.
Reviewer #3
- Introduction: author should add motivation and contribution in the introduction
- Conclusion: author should highlight the study limitation
- All: grammars and proofreading need improve
- Reference: some of the references need to update
Feedback for Reviewer #3
- Introduction: The introduction has been revised. The motivation and contribution of this research are in the introduction.
- Conclusion: The notes on limitations of this study have been revised and highlighted with a label in the conclusion.
- All: The manuscript has been extensively revised and checked.
- Reference: The reference list has been updated. The 2008 statistic about the knowledge economy from the United Nations has been updated with a recent reference, which cites the same data. In other areas of this research, historical and recent references provide an invaluable resource of information about the history and progress of development economics as well as the lived experiences between the Government of Belize and the World Bank Group.
Thanks again
Reviewer 4 Report (New Reviewer)
The authors argue that so-called developing countries see themselves and their status as needy, unable to develop themselves and in need of funds and knowledge.
This study is needed and well prepared, even ready to go to print without additional changes.
Author Response
Dear Sir or Madam:
On behalf of my co-writer, I would like to thank you for reviewing our first manuscript submission entitled: “Investigating Sustainable Human Development Funding of Belize in the World Bank Group: A Brand Relationship Theory Approach.” You gave valuable feedback, which has truly benefitted this extensively revised version.
After a request for a name change, it is now tentatively titled, “Arranged Marriages in Multilateral Partnerships - Investigating Sustainable Human Development Financing of Belize in the World Bank Group: A Brand Relationship Theory Approach.” Please find our feedback to your comments and suggestions below.
Reviewer #4
- Conclusion: The authors argue that so-called developing countries see themselves and their status as needy, unable to develop themselves and in need of funds and knowledge.
- All: This study is needed and well prepared, even ready to go to print without additional changes.
Feedback for Reviewer #4
- Conclusion: Thanks for pulling out this sentence from the conclusion. Some of it has been added to the abstract. Yes, this is the way member countries can see themselves, based on the dysfunctional brand relationship with the World Bank, it is very unhealthy and not indicative of a true partnership.
- All: Thank you so much for approving the original manuscript for publication. Based on feedback from other reviewers, the paper has been extensively revised to make it even better.
Thanks again
Round 2
Reviewer 2 Report (New Reviewer)
The revision has been carried out in full according to the comments submitted.
This manuscript is a resubmission of an earlier submission. The following is a list of the peer review reports and author responses from that submission.
Round 1
Reviewer 1 Report
Dear authors,
Thank you for the chance to read this paper. After reading it I think that it needs a different approach. In general, I think that it should focus on fewer concepts, and the relationship to be studied should be clearly presented. In my opinion it is not by now. Below you can find some comments that support my opinion.
Abstract:
According to the information available in the abstract, the reader might get confused. The objective is there, some research methodologies are presented, but it ends with a "request" to the world bank "to innovate its development economics practices and meet development needs of its members by building genuine 23 brand bonds." In a scientific paper, we should withdraw conclusion, not requests. Somehow, it takes us to an opinion article, or at much to a consultancy project. I suggest to the authors to improve the abstract, in particular the last part of it.
Introduction:
It starts with a brief overview and discussion about the paper subject, but then some questions arise about the research questions:
- What are the characteristics of the funding relationship in Belize’s membership in the World Bank Group? - simple and clear
- Based on the Government of Belize’s perceptions of the Bank, what is the brand relationship form [16] of Belize’s membership in the World Bank Group? - how will the authors get the government perceptions? The concept of "brand relationship" is not clear for me in this context? Who's brand? Belize or WB? I hope to get it clarified along the paper. The definition presented in line 120 "but in this paper a brand “has no objective existence at all: it is simply a collection of perceptions held in the mind of the consumer [client and customer] adds noise to this concept.
- How do these funding and brand relationships define challenges in meeting the country’s Sustainable Development Goals for human development? - Funding, brand relationships, challenges and sustainable goals for human development - all these concepts in the same research question?! it seems too much for one single research question
After the research questions, there is a brief overview of previous works that presented some suggestions about the relationship between Belize and the WB. Then some concepts clarifications. In this part and considering that the next chapter is the methodology, the authors should clarify their goals. The relationship between the WB and a country even with some similarities is different from a general supplier-buyer relationship. What are the brand aspects that are going to be explored? In this section the authors must clearly present the goals of the paper and clarify the concepts that are going to be analysed.
At this point it seems to be a concept mix: sustainable development, brand relationship, SDG, and human development.... By the end of the introduction I am not able to understand the goal of this paper.
3. Materials and Methods
I am not familiar with the adopted methodology, but it must be normal since the authors state in the abstract that this is a unique mixed-method approach. Without a clear vision of the paper goals, is difficult to evaluate the methodology.
Results:
No comments to do.
References:
Even with an interesting number of references, there is a large number of references related to news or associations such as UN or WB. In a total of 61 references, only 21 are scientific papers. In my opinion, this is a negative aspect, but justifies the low scientific approach of this paper.
Reviewer 2 Report
This paper represents a relevant contribution for investigating sustainable human development in terms of a brand relationship theory approach. The identified problem research in Belize itself seems reasonable, and certainly opens an avenue for discussion. The introduction to the paper nicely suggests that there still are gaps in knowledge that can be filled with the type of research conducted in this study. However, in my opinion some further considerations are the following:
- Irrespective of a particular model regarding brands, I suggest considering a general, integrative theoretical approach to present a conceptual framework previously, and then write the paper from the angle of the specific chosen approach.
- Even though the paper is empirically oriented, good papers (either theoretical or empirical) always provide a review of both types of papers related to their topic of study. The reasons for this are that different types of readers may be interested in reading the paper and more importantly it helps to better evaluate the merits of the paper’s contribution.
- In the literature review section, the information presented about different studies is unequal.
- The data subsection, discussions, and conclusion section are too brief, and the conclusions mentioned in the end of the article require more elaboration in previous sections. Particularly, a Conclusions Section regarding the ways in which this research with differing intervention contributes to managerial and theoretical implications in the study is required.
- Limitations of the study in terms of the generalization of the findings should be added.
Reviewer 3 Report
The paper examines the relationship between Belize and the World Banking group. The topic is actual in terms of whether the support of the World Bank Group provides a harmonious development between the different capital elements, thus serving the SD objectives, and whether they are truly appropriate to local needs. It is also important to what extent the quality of the cooperation serves to build trust in the institution, what is its brand. The Brand relation Theory Approach draws attention to the importance of trust and psychological capital.
The article in its current form is difficult to follow for international readers who are unfamiliar with Belize. It contains too much political information, using also gray publications, which is difficult to judge, but there are lacks of significant information. We do not know the level of economic development of the country and the contribution of the World Bank Group to the development, the lack of human capital investments and the reasons for it are not supported by data. The use of the Work Bank Brand for political purposes is an interesting proposition. The literature review lacks the results of empirical research in other developing countries on the effectiveness of the operation of the World Bank. Thus, most of the claims cannot be considered scientifically substantiated.
I miss the novel contribution and clear messages that the paper is able to offer to extant research for the global audience. Because of the missing prior research review the generalization of the results is not possible. At the same time, there is no doubt that attention to the need for a multifaceted analysis of the relationship between the Work Bank group and the cooperating countries is an outstanding merit of the article. For this reason, after a major revision , I suggest an opportunity for publication
Reviewer 4 Report
Dear Authors,
As it stands, the justification or novelty of the study remains a major shortcoming of this study. The Introduction makes a weak attempt to justify Belize as a representative case for all developing countries. The introduction of branding as a concept is very abrupt and arbitrary. For instance the following: “The literature points out a difference between the Banks’ branding and its real brand.”
The introduction should respond to the following: (1) Why is the research important in theory and practice? (2) What is the major unaddressed paradox or puzzle that this study seeks to resolve? (3) How does this study fundamentally change / challenge/advance scholars’ understanding of the multidisciplinary context of development economics and branding (marketing)? The paper moves back and forth and selectively applies concepts and theories without bothering to connect the dots. As a consequence, the paper is uneasy to read, verbose, and seems vaguely worded.
What do the authors mean by: “this is a mixed-methods, stand-alone literature review research. This researcher.” How is it mixed-methods? What is the meaning and significance of “stand-alone” literature review research?
The author(s) need to mention in detail how this study is distinct from other studies in the field. Also, explain the expected theoretical/practical utility based on each research question, and then towards the concluding section, you need to elaborate how (and if) you were able to find convincing answers to these questions. The conclusions section is poorly written. You may choose to re-write this section to properly reflect the implications of this study without meandering or rehashing the results. There has to be a strong focus on the implications of the study for other developing countries across different continents.
In the concluding section, the “limitations of the study” need to be elaborated. Overall, the paper could benefit from proofreading.
Best wishes.